# Palmitoyltransferase DHHC9 and acyl protein thioesterase APT1 modulate renal fibrosis through regulating β-catenin palmitoylation

Mengru Gu[1,2], Hanlu Jiang[1], Mengzhu Tan[1], Long Yu[1], Ning Xu[1], Ying Li[1], Han Wu[1], Qing Hou[1] & Chunsun Dai ⬡[1,2] ✉

palmitoylation, a reversible post-translational modification, is initiated by the DHHC family of palmitoyltransferases and reversed by several acyl protein thioesterases. However, the role and mechanisms for protein palmitoylation in renal fibrosis have not been elucidated. Here we show protein palmitoylation and DHHC9 were downregulated in the fibrotic kidneys of mouse models and chronic kidney disease (CKD) patients. Ablating DHHC9 in tubular cells aggravated, while inducing DHHC9 overexpression with adeno-DHHC9 transfection or iproniazid treatment protected against kidney fibrosis in male mouse models. Mechanistically, DHHC9 palmitoylated β-catenin, thereby promoted its ubiquitination and degradation. Additionally, acyl protein thioesterase 1 (APT1) was induced in the fibrotic kidneys, which depalmitoylated β-catenin, increased its abundance and nuclear translocation. Ablating tubular APT1 or inhibiting APT1 with ML348 markedly protected against unilateral ureter obstruction (UUO) or ischemia/reperfusion injury (IRI)-induced kidney fibrosis in male mice. This study reveals the regulatory mechanism of protein palmitoylation in kidney fibrosis.

Chronic kidney disease (CKD) is associated with irreversible nephron loss, end-stage renal disease, and/or premature death. Current therapies have limited efficacy and only delay disease progression[1]. Thus, there is a need to develop therapeutic approaches to stop or reverse CKD progression. Similar to the liver and adipose tissue, the kidney is one of the major organs that regulates lipid metabolism[2]. Dysregulation of tubular epithelial cells (TECs) exacerbates renal injury and predispose patients to CKD. Recent studies have demonstrated that TECs are involved in lipid metabolism[3]. In response to kidney injury, the ability of TECs to use fatty acid oxidation is impaired, leading to lipid accumulation. Lipids not only serve as energy supplies, but also have vital roles in several biochemical reactions by serving as

substrates for protein lipidation, a type of co-translational or post-translational modifications (PTMs)[4]. However, limited studies have examined the role and function of lipidation in kidneys.

Among all lipidation modifications, protein S-acylation is the only reversible modification and is dynamically catalyzed by a group of highly conserved Asp-His-His-Cys (DHHC) motif-containing palmitoyl S-acyltransferases (PATs) and reversed by several acyl protein thioesterases (APTs). As palmitate is the predominant fatty acid, protein S-acylation is commonly (albeit less accurately) called protein S-palmitoylation[5]. Previous studies have reported that S-palmitoylation is associated with various human diseases. In particular, the loss of function of DHHC9 is associated with X-linked

[1]Center for Kidney Diseases, the Second Affiliated Hospital of Nanjing Medical University; Nanjing, China, 210009262 North Zhongshan Road, Nanjing, Jiangsu, China. [2]Department of Clinical Genetics, the Second Affiliated Hospital of Nanjing Medical University; Nanjing, China, 210009262 North Zhongshan Road, Nanjing, Jiangsu, China. ✉e-mail: daichunsun@njmu.edu.cn

intellectual disability and epilepsy, while mutations in ZDHHC15 and ZDHHC8 are associated with X-linked mental retardation and schizophrenia, respectively[6–8]. In cases of pathogen infections, Cys141 and Cys498 residues on ACE2 are S-palmitoylated by DHHC3 and de-palmitoylated by APT1, which is critical for the membrane-targeting of ACE2 and its secretion via extracellular vesicles[9]. A few recent studies have evaluated the role of palmitoylation in kidney diseases. For example, protein palmitoylation is important for the localization and expression levels of PKD1, which plays an important role in autosomal dominant polycystic kidney disease[10]. Additionally, palmitoylation is crucial for lipid raft targeting of the renal DRD1[11]. However, the role and mechanism of palmitoylation in renal fibrosis have not been elucidated.

The Wnt/β-catenin pathway is an evolutionarily conserved pathway involved in renal development and repair[12]. In healthy adults, Wnt/β-catenin signaling is suppressed. However, Wnt/β-catenin signaling is reactivated in various types of CKD, including the unilateral ureteral obstruction (UUO), ischemia-reperfusion injury (IRI), angiotensin or adriamycin infusion, diabetic kidney disease, and age-related kidney dysfunction models[13]. The binding of Wnt to its plasma membrane receptors initiates a series of signaling events that results in the stabilization of β-catenin, which enters the nucleus to regulate the expression of its target genes by interacting with the T cell factor/lymphoid enhancer factor (TCF/LEF) family of transcription factors. Previous studies have reported that sustained activation of Wnt/β-catenin signaling promotes kidney fibrosis[14]. β-catenin is primarily upregulated in the tubular epithelium of fibrotic kidneys, suggesting that tubular cells are the primary targets of canonical Wnt signaling[15]. However, the role of β-catenin signaling in tubule responses to chronic injury has not been completely elucidated.

This study demonstrated that the downregulation of DHHC9-mediated palmitoylation in tubular cells of patients with CKD and animal CKD models exacerbates kidney fibrosis. Mechanistically, DHHC9 palmitoylates β-catenin and promotes its ubiquitination and degradation, leading to the alleviation of renal fibrosis. While APT1 de-palmitoylates β-catenin and enhances its stability, contributing to renal fibrosis. These findings will aid in the development of new CKD therapies.

## Results

### Downregulation of renal protein palmitoylation promotes renal fibrosis

To determine the role of DHHC-catalyzed palmitoylation in renal fibrosis, renal protein palmitoylation was examined using an acyl-biotin exchange (ABE) assay (to detect S-acylation) with bioorthogonal palmitic acid probes, followed by click chemistry (to detect S-palmitoylation). The renal tissues were enriched with palmitoylated proteins under physiological conditions, whereas the level of protein palmitoylation was markedly downregulated in TECs of mice after UUO (Fig. 1a) or IRI surgery (Fig. 1c). To visualize protein palmitoylation in the mouse models, mice were intraperitoneally injected with labeled palmitic acid. The level of protein palmitoylation was significantly downregulated in the fibrotic region (Fig. 1b, d).

To assess the role of downregulated protein palmitoylation in renal fibrosis, 2-bromopalmitate (2-BP; the most common and non-selective palmitoylation inhibitor) was intraperitoneally administered to mice. 2-BP markedly exacerbated UUO- or IRI-induced kidney fibrosis in mice. Periodic acid-Schiff (PAS), Masson's trichrome staining, and immunoblotting or immunofluorescence staining of fibronectin (FN) revealed that compared with those in the kidneys of control mice, the levels of tubular damage and fibrotic lesions were higher in the kidneys of 2-BP-treated mice (Fig. 1e–i). Similar results were obtained with the IRI-induced renal fibrosis model (Fig. 1j–n). The level of protein palmitoylation was decreased in cultured primary tubular epithelial cells (PTCs) treated with TGFβ1 (Supplementary

Fig. 1a). 2-BP promoted fibronectin (FN) production in PTCs (Supplementary Fig. 1b). These results indicate that downregulated protein palmitoylation in tubular cells aggravates extracellular matrix production and renal fibrosis.

### Downregulation of DHHC9 in tubular cells of patients and mouse models with CKD

The kidneys were enriched with PATs (Fig. 2a). In the UUO- or IRI-induced fibrotic kidneys, Zdhhc6 and Zdhhc9 were largely down-regulated (Fig. 2b), which was confirmed using quantitative real-time polymerase chain reaction (qRT-PCR) analysis (Fig. 2c). Most of the other PATs were upregulated in response to IRI-induced kidney injury (Supplementary Fig. 2a). Single-cell transcriptome profiling of the kidneys revealed that Zdhhc6 and Zdhhc9 were primarily expressed in renal podocytes and TECs, respectively (Fig. 2d). Furthermore, the expression of Zdhhc9 gene in TECs was downregulated after IRI (Fig. 2e) or UUO surgery (Supplementary Fig. 3a). Therefore, we speculated that DHHC9 reduction contributes to decreased protein palmitoylation in TECs during renal fibrosis. Immunohistochemical staining showed that DHHC9 protein is expressed in the distal and proximal tubule of the mice, which was decreased in kidneys from mice after UUO (Fig. 2f, g) or IRI surgery (Fig. 2h, i). Consistently, downregulation of DHHC9 in TECs was accompanied by markedly downregulated palmitoylation (Fig. 2j). Analysis of the kidney transcriptome database of patients with CKD revealed that the renal DHHC9 mRNA levels were reduced in CKD patients (Fig. 2k). In particular, the expression of DHHC9 was reduced in TECs of CKD patients, including diabetic nephropathy (DN), IgA nephropathy (IgAN), and membranous nephropathy (MN) (Fig. 2l). Thus, DHHC9 abundance was markedly downregulated in TECs from mouse and patients with CKD.

### DHHC9 inhibits tubular cell extracellular matrix production and kidney fibrosis

The role of DHHC9 in renal fibrosis was then evaluated. TGFβ1 time-dependently downregulated the expression of DHHC9 in PTCs (Fig. 3a). Short-interfering RNA (siRNA)-mediated knockdown of DHHC9 promoted (Fig. 3b), whereas the adenoviral vector-mediated overexpression of DHHC9 suppressed TGFβ1-induced FN production in TECs (Fig. 3c). To investigate the role of tubular cell DHHC9 downregulation in the progression of kidney fibrosis, a mouse model with DHHC9-deleted tubular cells was generated using the Cre/LoxP system (Supplementary Fig. 4a–c). The knockouts were born normal. No difference as to body weight, kidney weight, and urine β-N-Acetyl glucosaminidase (NAG) and blood urea nitrogen (BUN) was found between the knockouts and their control littermates at month 2 post-birth (Supplementary Fig. 4d–h). Tub-DHHC9$^{-/-}$ mice and control littermates were subjected to UUO. At day 10 post-surgery, Tub-DHHC9$^{-/-}$ mice exhibited exacerbated tubule atrophy, increased collagen deposition and FN production when compared with control littermates as evidenced by the results of PAS, Masson's trichrome staining, and FN immunofluorescence staining (Fig. 3d–f) and immunoblotting (Fig. 3g). Similarly, the deletion of tubular cell DHHC9 exacerbated IRI-induced kidney fibrosis in mice (Fig. 3h–k). Additionally, the adenoviral vector was used to overexpress DHHC9 in mouse kidneys by multisite in situ injection (Supplementary Fig. 5). After UUO, kidney fibrosis in the DHHC9 adenoviral vector transfected mice was ameliorated when compared with that in control mice (Fig. 3l, m and o). Similar results were obtained with the IRI-induced renal fibrosis model (Fig. 3l, n, and p). Taken together, these findings suggest that tubular cell DHHC9 protects against renal fibrosis.

### DHHC9 alleviates renal fibrosis by downregulating β-catenin expression

The mechanisms involved in the protective effects of DHHC9 against renal fibrosis were evaluated. The coordinated effects of Zdhhc9

knockout across the Cancer Dependency Map (www.depmap.org), a resource of genome-wide clustered regularly interspaced palindrome repeat (CRISPR)-caspase 9 knockout screens in hundreds of cancer cell lines, were examined. Genes exhibiting similar effects on viability, even if subtle, suggested that they have a shared function or

regulatory mechanism. The effects of β-catenin and *Zdhhc9* knockout were negatively correlated with cell survival (Fig. 4a). Previous studies have demonstrated that activating the Wnt/β-catenin signaling triggers a cascade of downstream events, leading to the β-catenin stabilization, nuclear translocation, and the upregulation

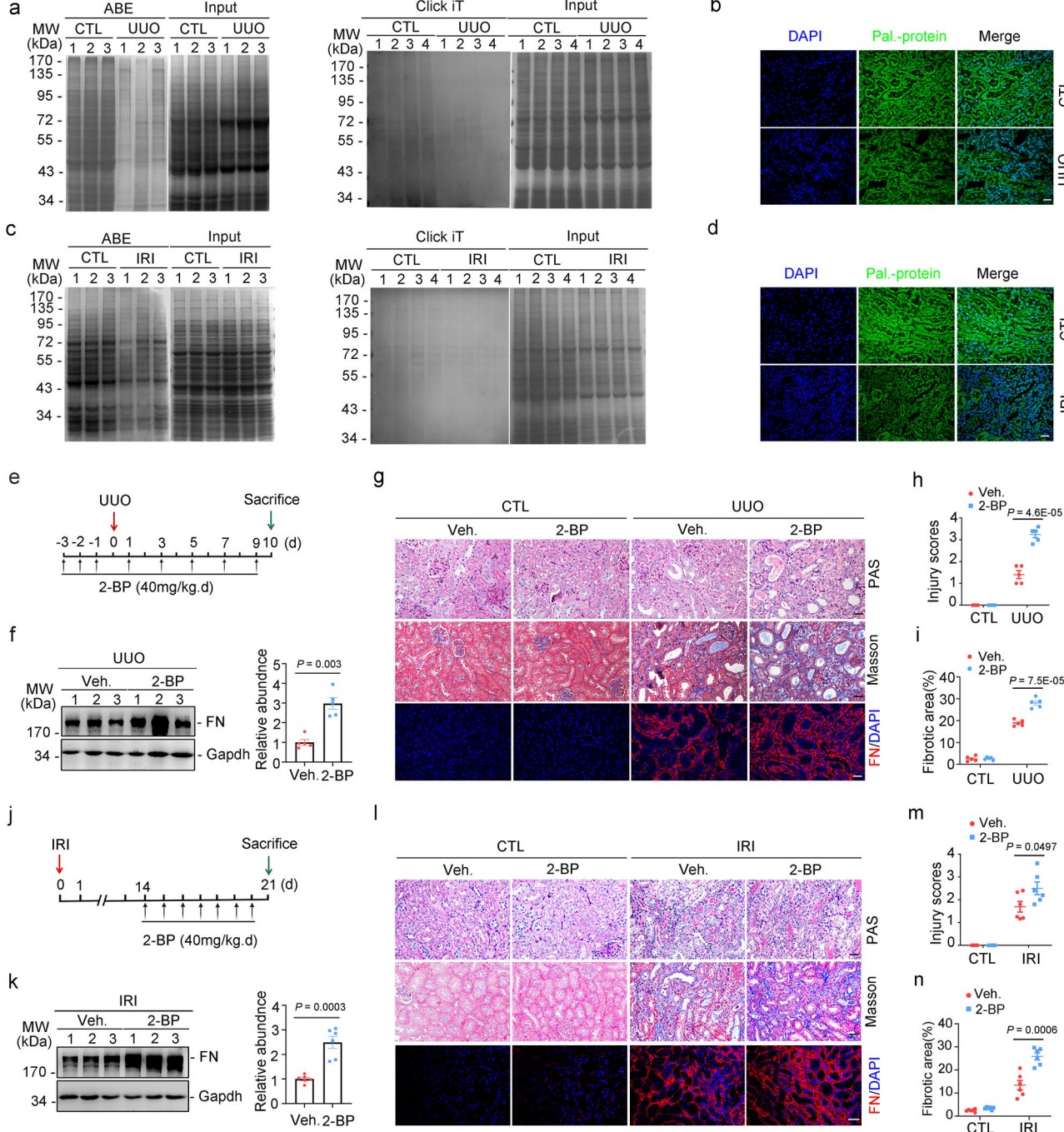

**Fig. 1 | Reduction of protein palmitoylation exacerbates kidney fibrosis.** ABE and Click-iT assay (**a**), and Click-iT chemistry fluorescence (**b**) showing the reduced protein palmitoylation in mouse kidneys after UUO (unilateral ureter obstruction). Pal-protein, palmitoylated protein. Scale bar, 20 μm. ABE and Click-iT assay (**c**), and Click-iT chemistry fluorescence (**d**) showing the reduced protein palmitoylation in mouse kidneys after IRI (ischemia/reperfusion injury). Scale bar, 20 μm. **e** Strategy for UUO surgery and 2-BP (2-bromopalmitate, the palmitoylation inhibitor) administration. **f** Western blot and quantitative analyses for FN (fibronectin) in UUO kidneys. *n* = 5 biologically independent animals. Representative images of periodic acid-Schiff (PAS), Masson-trichrome and FN staining in kidneys. Scale bar, 20 μm

(**g**), injury scores (**h**) and the fibrotic area (**i**) among groups as indicated. *n* = 5 biologically independent animals. **j** Strategy for IRI surgery and 2-BP administration. **k** Western blot and quantitative analyses for FN in IRI kidneys. *n* = 6 biologically independent animals. Representative images of PAS, Masson-trichrome and FN staining in kidneys. Scale bar, 20 μm (**l**), injury scores (**m**) and the fibrotic area (**n**) among groups as indicated. *n* = 6 biologically independent animals. Representative results were obtained from at least three independent experiments with similar results. Data represent the mean ± SEM. *P* values were determined by the two-tailed Student's *t*-test.

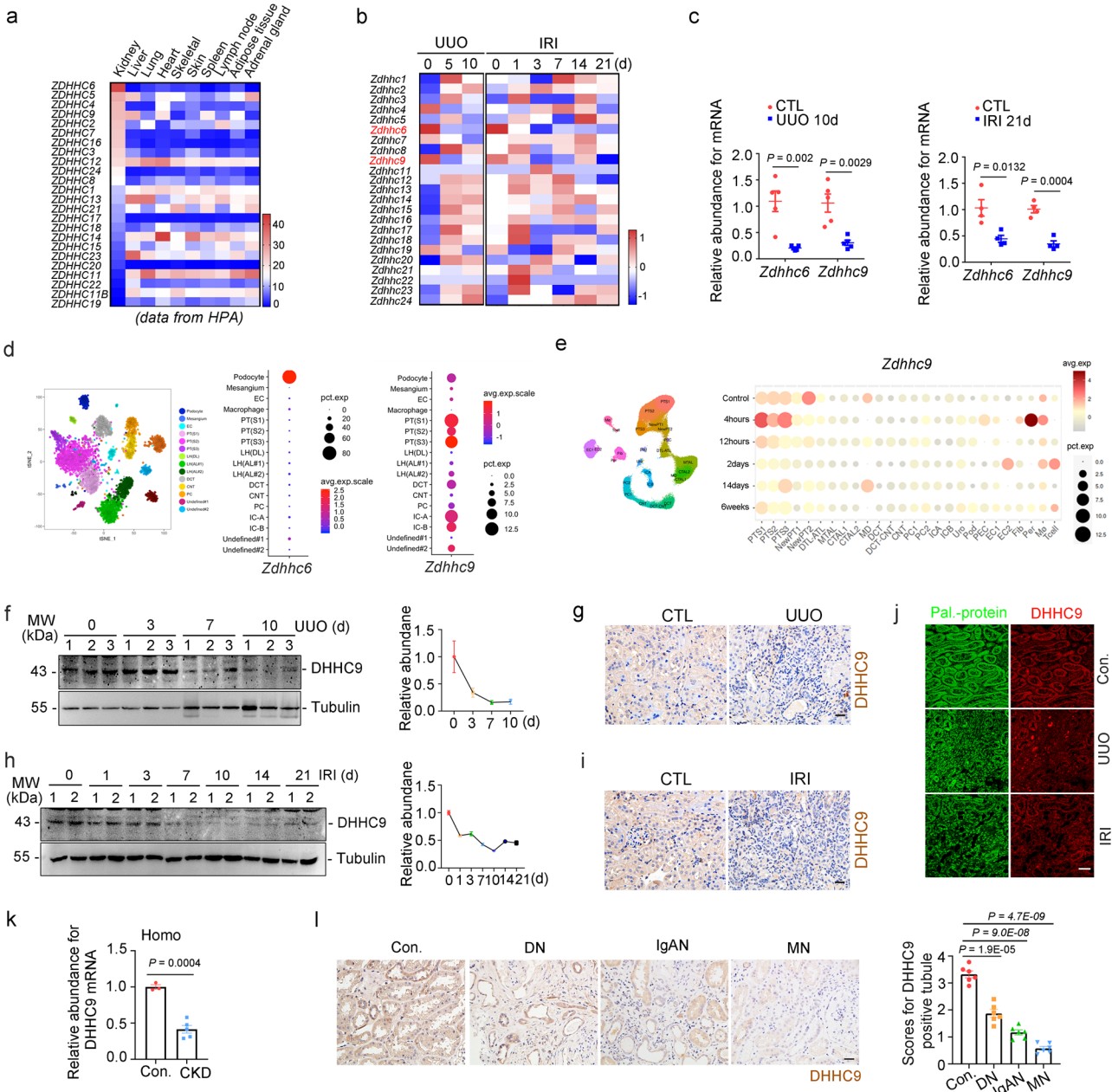

**Fig. 2 | Reduction of DHHC9 in tubule from mice and patients with CKD.** Heat map of *ZDHHCs* gene expression in different organs (From Human Protein Atlas, HPA) (**a**) and in mouse kidneys after UUO (GSE125015[47]) or IRI (GSE98622[43]) (**b**). **c** qRT-PCR analyses for *Zdhhc6* and *Zdhhc9* in mouse kidneys. *n* = 5. **d** The distribution and relative expression of *Zdhhc6* and *Zdhhc9* in mouse kidneys (From snRNA-Seq database[48]; http://humphreyslab.com/SingleCell/). **e** The distribution and relative expression of *Zdhhc9* in different renal cell types form mouse kidneys after IRI (From snRNA-Seq database[49]; http://humphreyslab.com/SingleCell/). Western blot and quantitative analyses and immumohistochemical staining for

DHHC9 in UUO kidneys (**f**, **g**) or IRI kidneys (**h**, **i**). *n* = 3. Scale bar, 20 μm. **j** Click-iT chemistry fluorescence and co-staining of DHHC9 in kidneys. Scale bar, 20 μm. **k** The *Zdhhc9* mRNA expression in CKD patients (From GSE69438[50]). *n* = 3. **l** Representative immunohistochemical staining images of DHHC9 and quantitative analyses in tubule from CKD patients including diagnosis of Diabetic nephropathy (DN), IgA nephropathy (IgAN) and Membranous nephropathy (MN). *n* = 6. Scale bar, 20 μm. Representative results were obtained from at least three independent experiments with similar results. Data represent the mean ± SEM. *P* values were determined by the two-tailed Student's *t*-test.

of the target gene expression[16]. We hypothesized that *Zdhhc9* knockout promotes renal fibrosis by upregulating β-catenin expression. 2-BP upregulated the levels of β-catenin and its active form and downstream target protein (Cyclin D1) in PTCs but downregulated the levels of phosphorylated β-catenin (inactivated form, phosphorylated at serine 33 and 37 sites) in a dose-dependent and time-dependent manner (Fig. 4b, c). The siRNA-mediated downregulation of DHHC9 upregulated the expression of β-catenin and its active form and Cyclin D1, but suppressed the phosphorylated β-catenin (Fig. 4d). DHHC9 knockdown not only increased β-catenin nuclear

translocation (Fig. 4e) but also upregulated its activity as a transcription factor in PTCs (Fig. 4f). Consistent results were obtained in PTCs transfected with DHHC9 adenovirus (Fig. 4g–i). TGFβ1 upregulated β-catenin expression and nuclear translocation, as well as its binding to TCF/LEF, whereas DHHC9 overexpression exerted contrasting effects in PTCs (Supplementary Fig. 6a and Fig. 4h, i). In mice, compared with those in control littermates, the renal tubular levels of β-catenin were upregulated in Tub-DHHC9$^{-/-}$ and Tub-DHHC9$^{-/-}$ mice after UUO or IRI surgery (Fig. 4j–l). Immunofluorescence staining revealed that the tubular levels of DHHC9 were

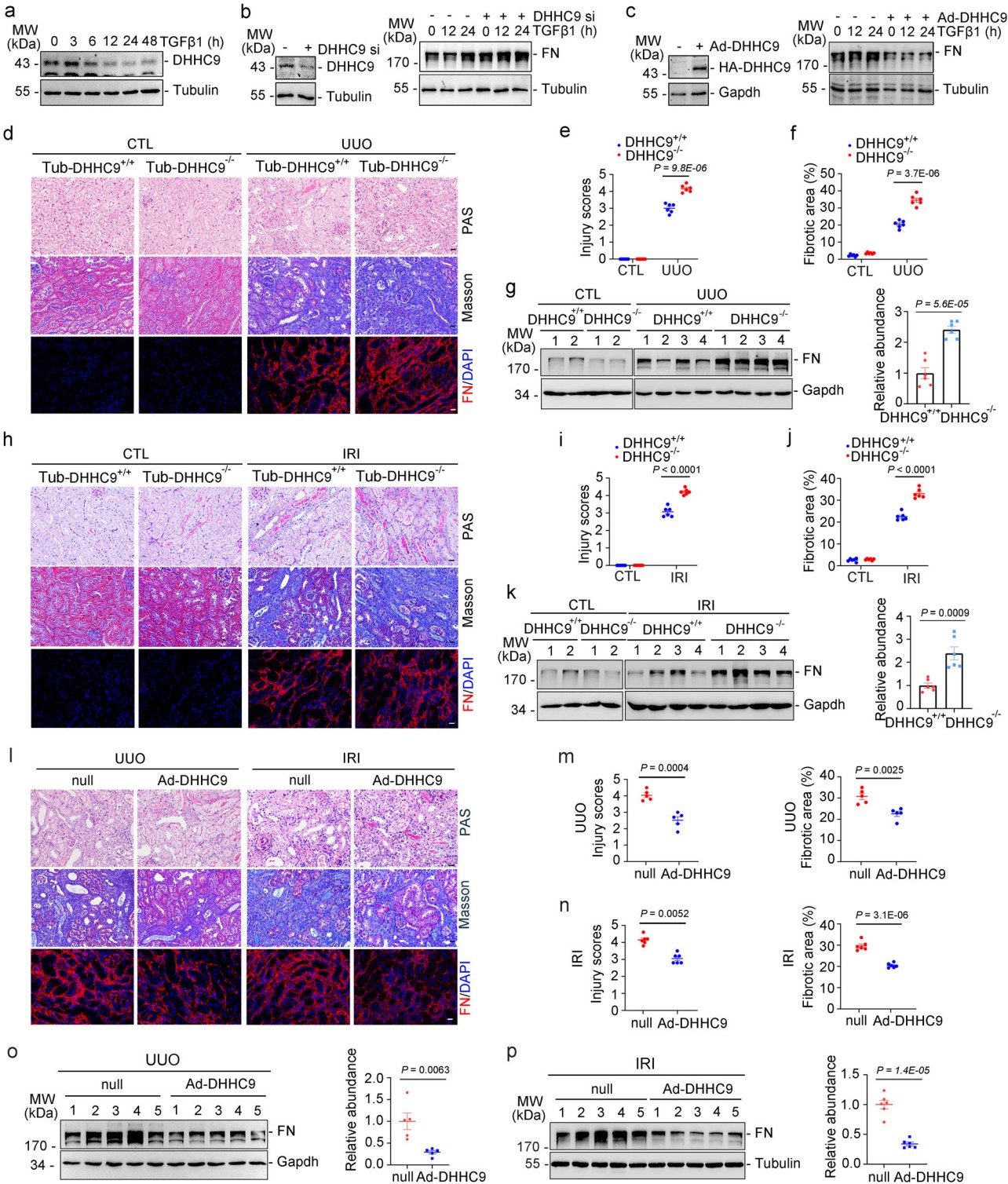

**Fig. 3 | DHHC9 inhibits tubular cell-extracellular matrix production and kidney fibrosis. a** Western blot analyses showing DHHC9 downregulation in PTCs treated with TGFβ1. Western blot analyses showing DHHC9 knocking down promotes (**b**), while DHHC9 overexpression inhibits (**c**) FN production in PTCs. Representative images of PAS, Masson-trichrome and FN staining in kidneys after UUO. Scale bar, 20 μm (**d**), injury scores (**e**) and the fibrotic area (**f**). *n* = 6. **g** Western blot and quantitative analyses showing the FN abundance in kidneys among groups. *n* = 6. Representative images of PAS, Masson-trichrome and FN staining in kidneys after IRI. Scale bar, 20 μm (**h**), injury scores (**i**) and the fibrotic area (**j**). *n* = 6. **k** Western

blot and quantitative analyses for FN in kidneys. *n* = 6. Representative images of PAS, Masson-trichrome and FN staining in kidneys after UUO or IRI surgery. Scale bar, 20 μm (**l**), injury scores and the fibrotic area in kidneys after UUO (**m**) or IRI (**n**). *n* = 5 biologically independent animals. Western blot and quantitative analyses for FN in kidneys after UUO (**o**) or IRI (**p**). *n* = 5 biologically independent animals. Representative results were obtained from at least three independent experiments with similar results. Data represent the mean ± SEM. *P* values were determined by the two-tailed Student's *t*-test.

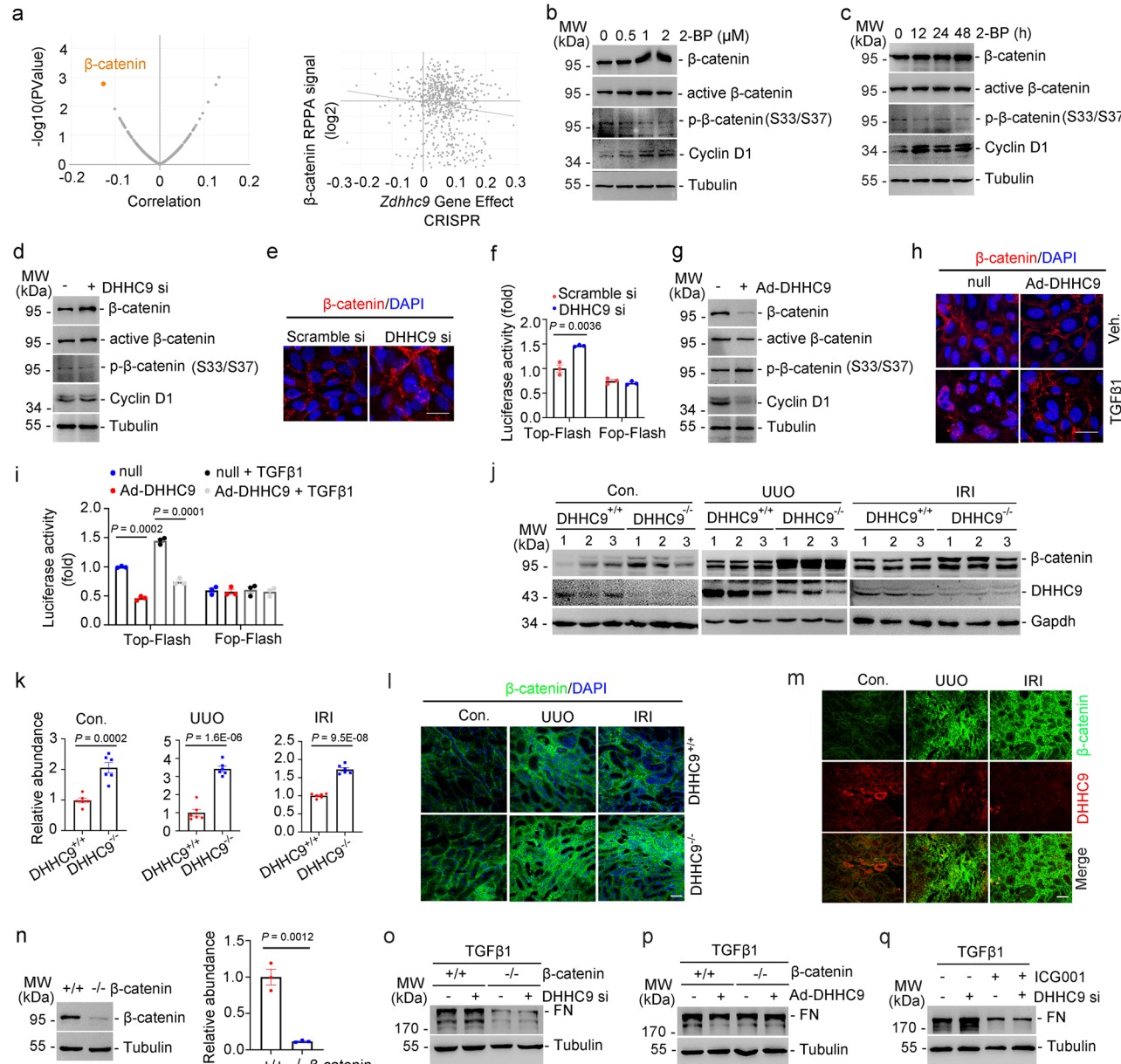

**Fig. 4 | DHHC9 inhibits the extracellular matrix production by promoting β-catenin degradation in tubular cells. a** Correlation analysis of gene dependencies identifying β-catenin as the target of DHHC9 (From the Cancer Dependency Map). Western blot analyses for β-catenin, active form, phosphorylated status, and Cyclin D1 in PTCs treated with 2-BP at different doses (**b**) or time (**c**). Western blot analyses for β-catenin, active form, phosphorylated status, and Cyclin D1 (**d**), and immunofluorescence staining for β-catenin (**e**) and TOP/FOP-flash luciferase activity (**f**) in PTCs with DHHC9 siRNA transfection. Scale bar, 10 μm. *n* = 3. Western blot analyses for β-catenin, active form, phosphorylated status, and Cyclin D1 (**g**), and immunofluorescence staining for β-catenin (**h**), and TOP/FOP-flash luciferase activity (**i**) in PTCs with DHHC9 adenovirus transfection. Scale bar, 10 μm. *n* = 3. Western blot analyses (**j**) and relative protein abundance (**k**), and representative immunofluorescence staining images for β-catenin (**l**) in kidneys. *n* = 6. Scale bar, 20 μm.

**m** Representative immunofluorescence staining images for β-catenin and DHHC9 in kidneys among groups as indicated. Scale bar, 20 μm. **n** Western blot analyses and quantitative analyses showing the β-catenin expression in the control (β-catenin⁺/⁺) and β-catenin-ablated (β-catenin⁻/⁻) PTCs. *n* = 3. **o** Western blot analyses showing β-catenin ablation inhibiting fibronectin production in DHHC9-knocked down PTCs with TGF-β1 treatment. **p** Western blot analyses showing the effect of DHHC9 overexpression on fibronectin production in the control (β-catenin⁺/⁺) and β-catenin-ablated (β-catenin⁻/⁻) PTCs. **q** Western blot analyses showing ICG001 treatment inhibiting fibronectin production in DHHC9 knocked down PTCs with TGF-β1 treatment. Representative results were obtained from at least three independent experiments with similar results. Data represent the mean ± SEM. *P* values were determined by the two-tailed Student's *t*-test.

downregulated, whereas those of β-catenin were upregulated in mice after UUO or IRI surgery (Fig. 4m).

To evaluate the effect of DHHC9-mediated regulation of β-catenin on extracellular matrix production in PTCs, β-catenin-deleted PTCs were generated by infecting β-catenin-floxed PTCs with adenovirus-Cre (Fig. 4n). β-catenin deletion significantly suppressed the TGFβ1-induced FN production in PTCs. The effect of DHHC9 knockdown on

FN production was markedly suppressed in β-catenin-deleted PTCs (Fig. 4o). Meanwhile, DHHC9 overexpression did not downregulate FN production in β-catenin-deleted PTCs (Fig. 4p). Additionally, PTCs were treated with ICG001, a small molecule that specifically inhibits TCF/β-catenin transcription in a cAMP response element-binding protein-dependent manner[17]. 2-BP (Supplementary Fig. 6b) or DHHC9 knockdown (Fig. 4q) did not upregulate FN production in PTCs

treated with ICG001. Moreover, a mouse model of tubular cell β-catenin-deletion was established (Supplementary Fig. 7a). UUO or IRI-induced kidney fibrosis in Tub-β-catenin⁻/⁻ mice was ameliorated when compared with that in control littermates as evidenced by downregulated FN production, decreased tubule damage and collagen deposition (Supplementary Fig. 7b–g). Together, these results indicate that DHHC9 alleviates renal fibrosis by downregulating β-catenin levels.

Screening for drugs that suppress the effects of *Zdhhc9* knockout on cell survival across the Cancer Dependency Map and considering the generality between cell lines and the reproducibility of effects, iproniazid was selected as a drug to induce DHHC9 expression (Supplementary Fig. 8a). Treatment with 200 nM iproniazid markedly upregulated DHHC9 expression and downregulated β-catenin levels in PTCs (Supplementary Fig. 8b, c). Moreover, iproniazid obviously antagonized TGFβ1-induced FN production and blocked DHHC9 knock down-promoted FN production in PTCs treated with TGFβ1 (Supplementary Fig. 8d). Animal studies have demonstrated that iproniazid upregulated DHHC9 expression and downregulated β-catenin expression in the kidneys after UUO surgery (Supplementary Fig. 8e–g). Additionally, iproniazid treatment alleviated UUO-induced renal fibrosis at a dose of 0.3 mg/kg body weight/day (Supplementary Fig. 8f–j). Similar results were obtained with the IRI-induced renal fibrosis model (Supplementary Fig. 8k–p). Thus, iproniazid upregulated DHHC9 expression to protect the kidney against UUO- or IRI-induced fibrosis.

## DHHC9 palmitoylates β-catenin at cysteine 300 and promotes its ubiquitination and subsequent degradation

The mechanism involved in DHHC9-mediated regulation of β-catenin levels was examined. As DHHC9 is a palmitoyltransferase, we speculated that DHHC9 reduces β-catenin abundance by regulating its palmitoylation. Hence, the role of palmitoyl-CoA in modifying β-catenin was examined. The ABE assay showed that β-catenin undergoes S-acylation (Fig. 5a). Then, the type of acyl-CoA substrate required to modify β-catenin was determined. The results of a bioorthogonal palmitic acid probes assay revealed that palmitoyl-CoA (C16) was the preferred substrate for β-catenin (Fig. 5b). Further, the role of DHHC9 in β-catenin palmitoylation was examined. The level of palmitoylated β-catenin was consistent with that of DHHC9 in PTCs treated with TGFβ1 (Fig. 5c). Immunoprecipitation analyses revealed that DHHC9 and β-catenin form complexes in PTCs (Fig. 5d). Additionally, immunofluorescent analyses revealed the colocalization of hemagglutinin (HA)-tagged DHHC9 with β-catenin, as well as the colocalization of endogenous DHHC9 with green fluorescent protein (GFP)-tagged β-catenin in PTCs (Supplementary Fig. 9a, b). The purified DHHC9 could directly bind to the purified β-catenin in an in vitro binding assay (Supplementary Fig. 9c). ABE assays revealed that DHHC9 knockdown decreased the level of palmitoylated β-catenin. In contrast, DHHC9 overexpression increased the level of palmitoylated β-catenin (Fig. 5e, f). An in vitro PAT assay was performed using purified recombinant ZDHHC9 and β-catenin in the presence of palmitoyl azide-CoA as the palmitate donor. The terminal azide group of this compound allows the palmitoylation of proteins to be assessed using the click chemistry linking reactions (Fig. 5g). Wild-type (WT) DHHC9, but not catalytically inactive DHHC9 C169S mutant, significantly increased β-catenin palmitoylation (Supplementary Fig. 9d). Therefore, these results suggest that DHHC9 palmitoylates β-catenin.

Next, the palmitoylation sites of β-catenin were examined. APE assay demonstrated that β-catenin has multiple palmitoylated sites (Supplementary Fig. 9e). Further, we combined the predicted palmitoylation sites from CSS-Palm 4.0 and GPS-Palm software programs and filtered out 6 sites including C300, C381, C466, C520, C573 and C619 (Supplementary Fig. 9f). To determine the primary site of β-catenin palmitoylation, we mutated each cysteine to serine or all six

cysteines to serines in β-catenin. The S-palmitoylation levels were characterized using the azide palmitic acid incorporation assay. The C300S mutation and the mutations in six cysteine residues nearly abolished the palmitoylation of β-catenin (Fig. 5h and Supplementary Fig. 9g), suggesting that Cys 300 is the major site for β-catenin palmitoylation. The level of C300S mutant was higher than that of WT β-catenin (Fig. 5h, bottom). Additionally, the level of C300S in complex with TCF/LEF was upregulated when compared with WT β-catenin (Supplementary Fig. 9h). The results of azide palmitic acid incorporation assay revealed that silencing DHHC9 expression downregulated the palmitoylation of WT β-catenin, but not that of C300S mutant (Fig. 5i, upper). Consistent results were obtained in PTCs with DHHC9 overexpression (Fig. 5i, lower). Moreover, the results of the in vitro PAT assay revealed that DHHC9 could catalyze the palmitoylation of WT β-catenin, but not that of C300S mutant (Fig. 5j). These findings indicate that DHHC9 palmitoylates β-catenin at cysteine 300.

Next, the mechanism underlying the DHHC9-mediated downregulation of β-catenin levels was examined. Palmitoylation changes the hydrophobicity of the protein and affects its binding to individual organelle membranes. Thus, the distribution of β-catenin in PTCs was analyzed. The results of experiments involving the downregulation or upregulation of DHHC9 were consistent with those of experiments with β-catenin alterations in the cell membrane, cytosol, and nucleus, indicating that DHHC9 did not affect the directional movement of β-catenin in cell organelles (Supplementary Fig. 9i). Furthermore, knocking down DHHC9 expression did not upregulate the transcriptional level of β-catenin (Supplementary Fig. 9j). Meanwhile, the proteasome inhibitor lactacystin significantly increased both endogenous (Fig. 5k) and exogenous β-catenin (Supplementary Fig. 9l) abundance in DHHC9-overexpressed PTCs. In addition, a β-catenin N terminal, which is required for β-catenin degradation[18], deletion plasmid was constructed. DHHC9 overexpression decreased WT β-catenin abundance, but not those of N terminal-deleted β-catenin (Supplementary Fig. 9k). These data suggest that DHHC9 regulates β-catenin proteasome degradation. Next, we examined the mechanism of DHHC9 on β-catenin degradation. Silencing DHHC9 expression reduced β-catenin ubiquitination and its binding to Axin, casein kinase (CK) 1α and glycogen synthase kinase (GSK) 3β, which are members of the β-catenin degradation complex (Fig. 5l, m, left). Consistent results were obtained with DHHC9 overexpression (Fig. 5l, m, right). The level of β-catenin C300S ubiquitination and binding to Axin or GSK3β were markedly reduced when compared with those of WT β-catenin (Fig. 5n). The half-life of exogenous β-catenin C300S was significantly higher than that of WT β-catenin in PTCs treated with cycloheximide (CHX), a translational inhibitor (Fig. 5o). Additionally, the distribution of exogenous β-catenin C300S was increased in the cellular membrane and nucleus (Supplementary Fig. 9m). These results suggest that DHHC9 catalyzed the palmitoylation of β-catenin at cysteine 300 and promoted its ubiquitination and degradation.

Finally, the role of cysteine 300-palmitoylated β-catenin in renal fibrosis was explored. FN production in β-catenin C300S-overexpressing and TGFβ1-treated PTCs was more than that in WT β-catenin-overexpressing PTCs (Supplementary Fig. 9n). Compared to WT β-catenin, C300S mutant exhibited increased levels of expression and nuclear translocation upon treatment with or without TGFβ1 (Supplementary Fig. 9O). In addition, we performed in situ injections of adeno-associated virus (AAV) containing β-catenin WT or C300S in mouse kidneys (Supplementary Fig. 10a, g). We found that C300S AAV injected mice (Supplementary Fig. 10b, h) had more severe renal fibrosis compared to WT β-catenin injected mice after UUO or IRI surgery (Supplementary Fig. 10c–f, i–l). These findings indicate that DHHC9 catalyzes the palmitoylation of β-catenin at the C300 site and, promotes its degradation, and inhibits FN production and renal fibrosis.

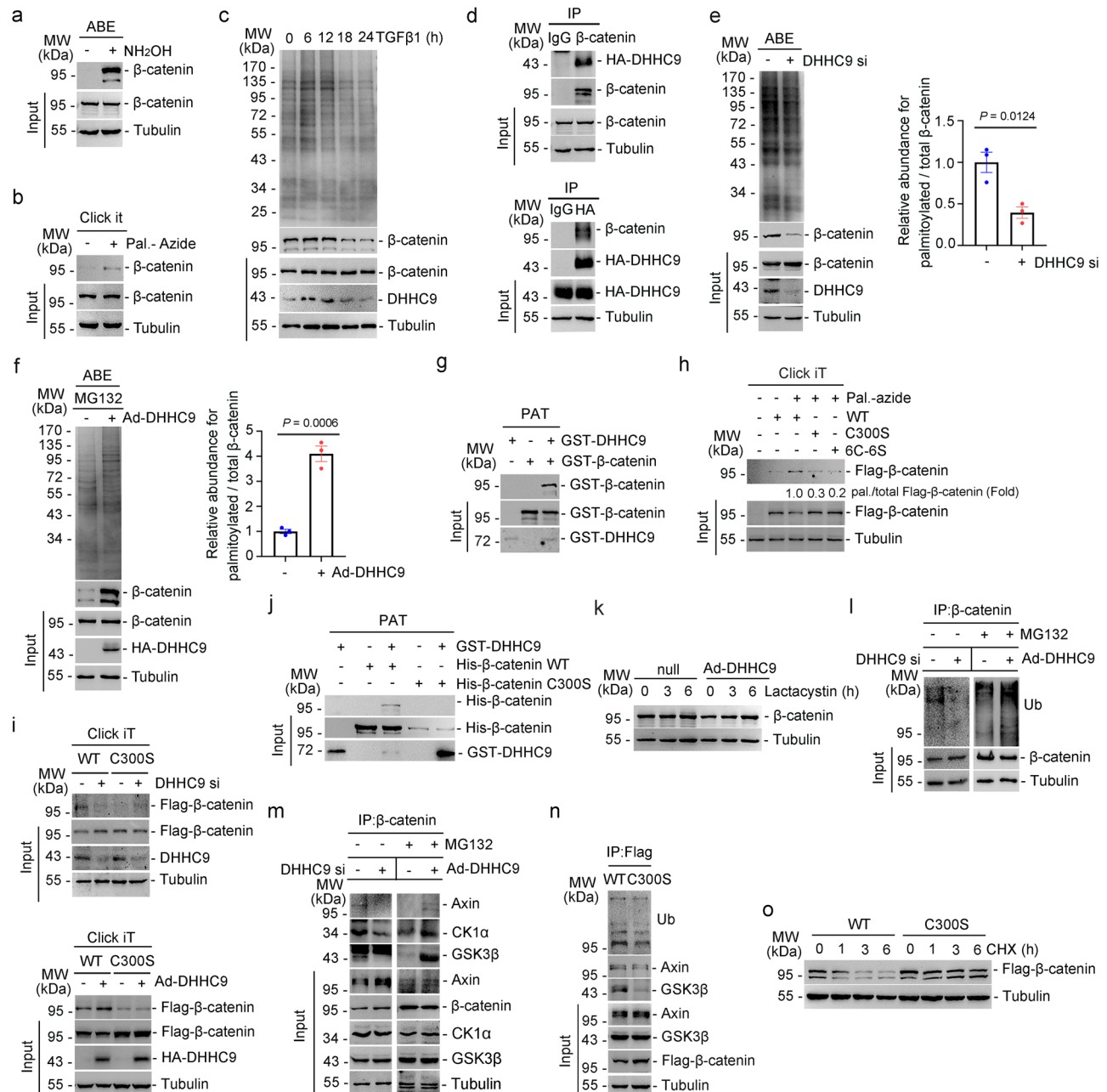

**Fig. 5 | DHHC9 palmitoylates β-catenin at Cys 300 to facilitate its degradation.**
ABE (**a**) and Click-iT (**b**) assay showing β-catenin palmitoylation in PTCs. Pal.-Azide, Azide labeled palmitic acid. **c** ABE assay showing the reduced total protein and β-catenin palmitoylation in TGFβ1 treated PTCs. **d** Immunoprecipitation (IP) assay showing β-catenin interacting with exogenous DHHC9 in HEK 293 A cells after treatment with lactacystin at 20 mM for 6 hours. ABE assay and quantitative analyses showing total protein and palmitoylated β-catenin in PTCs with DHHC9 knocking down (**e**) or DHHC9-overexpressed PTCs treated with MG132 (**f**). n = 3 biologically independent experiments. **g** In vitro palmitoylation analysis was performed by mixing purified ZDHHC9 protein with purified β-catenin protein in the presence of palmitoyl azide-CoA showing DHHC9 directly palmitoylted β-catenin. **h** Click-iT assay and semi-quantitative analyses showing β-catenin palmitoylation status in HEK 293A cells transfected with Flag-β-catenin or its mutants as indicated. **i** Flag-tagged WT β-catenin or C300S β-catenin were transfected in DHHC9-knocked down PTCs (upper) or in DHHC9-overexpressed PTCs (bottom). Western

blot analyses showing the levels of Flag-β-catenin and its palmitoylation in groups as indicated. **j** In vitro palmitoylation analysis was performed by mixing purified WT DHHC9 with purified WT β-catenin or β-catenin C300S in the presence of palmitoyl azide-CoA. The palmitoylation levels of β-catenin were analyzed by click it chemistry assays. **k** Western blot analysis for β-catenin in PTCs with DHHC9 overexpression. **l** β-catenin ubiquitination in PTCs transfected with DHHC9 siRNA (left) or adenovirus (right). **m** The interaction of β-catenin with the degradation complex components in PTCs transfected with DHHC9 siRNA or adenovirus. **n** β-catenin ubiquitination and the interaction of Flag-tagged β-catenin with the degradation complex components in PTCs. **o** Western blot analyses showing the stability of WT or C300S mutant β-catenin. Cells were treated with CHX (Cycloheximide, 100 μg/ml) for 0, 1, 3 and 6 h, and then collected and lysed for immunoblot. Representative results were obtained from at least three independent experiments with similar results. Data represent the mean ± SEM. P values were determined by the two-tailed Student's t-test.

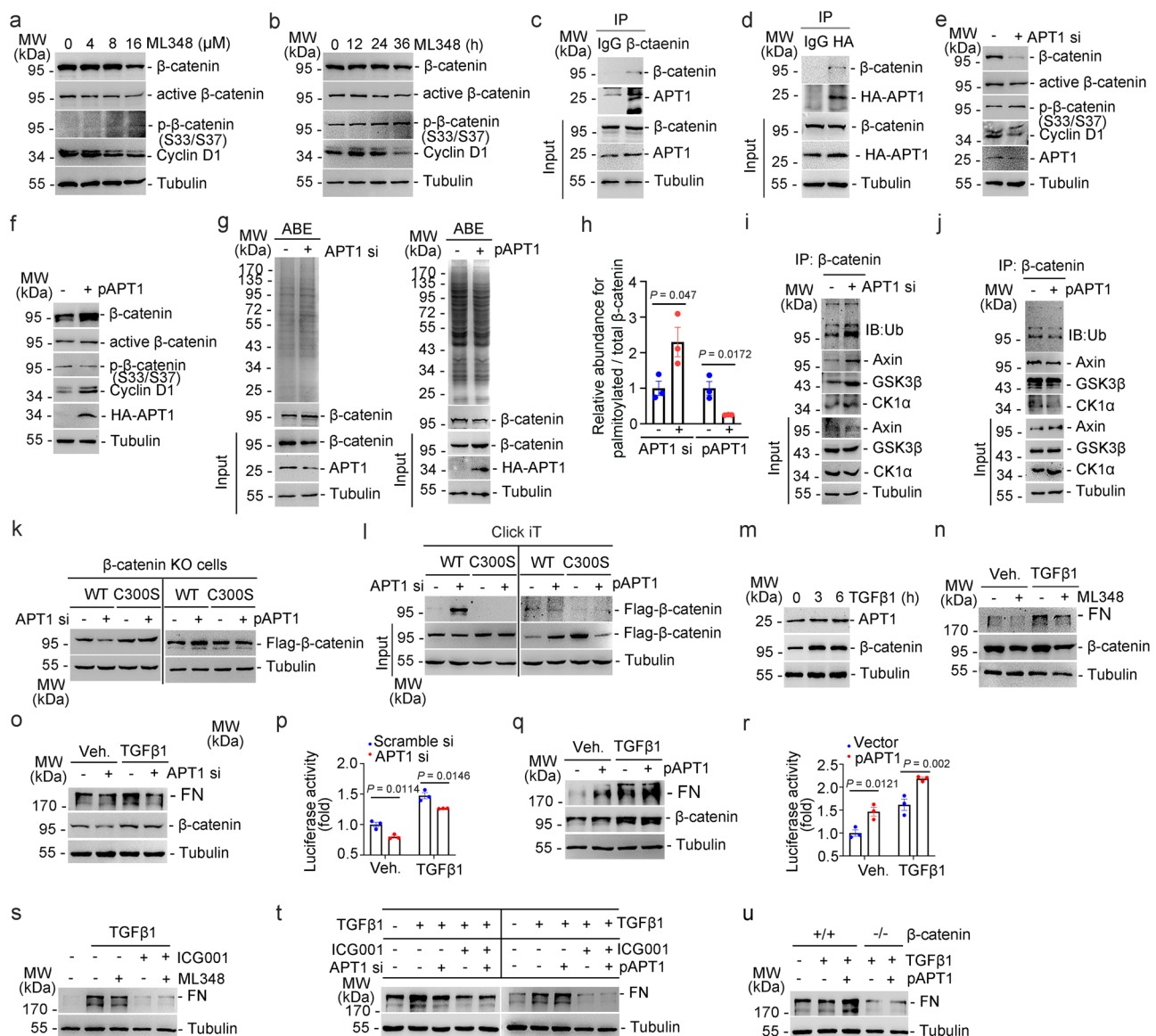

**Fig. 6 | APT1 depalmitoylates β-catenin to increase its stability and extracellular matrix production in tubular cells.** Western blot analyses showing total β-catenin levels, its active form and phosphorylated status, as well as Cyclin D1 abundance in PTCs treated with ML348 at different doses (**a**) or time (**b**). β-catenin interacting with endogenous (**c**) or exogenous (**d**) APT1 in PTCs. Western blot analyses showing total β-catenin levels, its active form and phosphorylated status, as well as Cyclin D1 abundance in PTCs transfected with APT1 siRNA (**e**) or APT1 plasmid (**f**). ABE assay (**g**) and quantitative analyses (**h**) showing total protein and palmitoylated β-catenin in PTCs with APT1 knocking down or plasmids transfection. n = 3 biologically independent experiments. The β-catenin ubiquitination and the interaction of β-catenin with the destruction complex components in PTCs transfected with APT1 siRNA (**i**) or APT1 plasmid (**j**). **k** Flag-tagged WT β-catenin or C300S β-catenin were transfected in β-catenin-ablated PTCs co-transfected with

APT1 siRNA (left) or APT1 plasmids (right). **l** Click it assay showing β-catenin palmitoylation. **m** Western blot analyses showing the induction of APT1 and β-catenin in PTCs treated with TGFβ1 (2 ng/ml). Western blot analyses showing the abundance of FN and β-catenin in PTCs treated with ML348 (**n**), in PTCs transfected with APT1 siRNA (**o**) or with APT1 plasmid (**q**). TOP/FOP-flash luciferase activity in PTCs transfected with APT1 siRNA (**p**) or APT1 plasmid (**r**). n = 3 biologically independent samples. Western blot analyses showing the effect of ICG001 treatment on fibronectin production in ML348-treated (**s**) or APT1 konck down (**t, left**) or APT1 overexpressed (**t, right**) PTCs with TGFβ1 treatment. **u** Western blot analyses showing FN abundance in PTCs as indicated. Representative results were obtained from at least three independent experiments with similar results. Data represent the mean ± SEM. P values were determined by the two-tailed Student's t-test.

## APT1 de-palmitoylates and stabilizes β-catenin to promote FN production in tubular cells

S-palmitoylation is a reversible lipid modification in which APTs catalyze β-catenin depalmitoylation. Two known cytosolic acyl protein thioesterases, APT1 and APT2 (also called LYPLA1 and LYPLA2), are thought to be responsible for depalmitoylating many S-acylated proteins[4]. Data from SwissPalm, a Protein Palmitoylation database, suggest that APT1 may depalmitoylate β-catenin[19]. ML348, a specific APT1 inhibitor, decreased β-catenin and its active form, and Cyclin D1 abundance, and increased its phosphorylated form (Fig. 6a, b). In

contrast, ML349, a specific APT2 inhibitor, did not affect the levels of β-catenin and its downstream target protein (Supplementary Fig. 11a, b). Immunoprecipitation assay showed that APT1 bind to β-catenin in PTCs (Fig. 6c, d). Immunofluorescent assay revealed the co-localization of GFP-tagged β-catenin with endogenous APT1, as well as the colocalization of exogenous GFP-tagged β-catenin with HA-tagged APT1 in PTCs (Supplementary Fig. 12a, b). Knocking down APT1 decreased the levels of β-catenin and its active form and Cyclin D1, and increased the level of phosphorylated β-catenin (Fig. 6e). The APT1 overexpression yielded consistent results (Fig. 6f). The results of ABE

assay revealed that silencing APT1 expression effectively increased (Fig. 6g, h, left), whereas overexpressing APT1 decreased β-catenin palmitoylation (Fig. 6g, h, right). Knocking down APT1 promoted β-catenin ubiquitination and its binding to Axin, CK1 and GSK3β (Fig. 6i), while overexpressing APT1 inhibited that (Fig. 6j). Additionally, silencing APT1 markedly decreased while overexpressing APT1 increased the level of WT β-catenin but not C300S mutant in β-catenin-ablated PTCs (Fig. 6k). The results of the click chemistry assays revealed that APT1 knockdown upregulated the level of WT β-catenin palmitoylation, but not that of C300S. In contrast, APT1 over-expression downregulated WT β-catenin palmitoylation (Fig. 6l). These results suggest that APT1 catalyzes β-catenin depalmitoylation at Cys300. Therefore, APT1 may be involved in extracellular matrix production by de-palmitoylating β-catenin. To prove that, TGFβ1 increased the levels of APT1 and β-catenin in PTCs (Fig. 6m). Inhibiting APT1 activity using ML348 or APT1 siRNA transfection effectively decreased TGFβ1-induced FN production in PTCs (Fig. 6n, o). APT1 knocking down inhibited β-catenin activity in PTCs with or without TGFβ1 treatment (Fig. 6p). Consistent results were obtained with APT1-overexpressing PTCs (Fig. 6q, r). Furthermore, the inhibition of ML348 on the extracellular matrix production was abolished in ICG001-treated PTCs (Fig. 6s). In PTCs treated with ICG001, APT1 silencing and APT1 overexpression did not affect TGFβ1-induced FN production (Fig. 6t). Similarly, in PTCs with β-catenin deletion, APT1 over-expression could not further upregulate TGFβ1-induced FN production (Fig. 6u). These findings indicate that APT1 de-palmitoylates β-catenin at Cys 300, inhibits its ubiquitination and degradation, and consequently promotes extracellular matrix production in TECs.

## Ablation or pharmacologically inhibition of APT1 in tubular cells alleviates renal fibrosis

In mice, the expression levels of APT1 and β-catenin were upregulated in kidneys after UUO (Fig. 7a) or IRI surgery (Fig. 7b). Immuno-fluorescence analysis revealed the induction and colocalization of APT1 and β-catenin in TECs of fibrotic kidneys (Fig. 7c). ML348 injection markedly attenuated interstitial matrix deposition and tubular atrophy in mice after UUO (Fig. 7d–i). We also performed IRI surgery and obtained similar results (Fig. 7j–o).

To further investigate the role of tubular APT1 induction in kidney fibrosis, we generated a mouse model with tubular cell APT1 ablation (Fig. 7p). Tub-APT1$^{-/-}$ mice showed less β-catenin or renal fibrosis after UUO or IRI, compared with control littermates (Fig. 7q–s). APT1 expression was upregulated in the tubular cells of patients with CKD, including those with DN, IgAN, and MN (Fig. 7t). Thus, APT1 is induced in TECs of animal models and patients with CKD. Targeting APT1 in tubular cells can mitigate renal fibrosis.

## Discussion

S-palmitoylation has piqued the interest of the scientific community in the field of PTM biology because of its reversible, enzyme-driven nature and wide-ranging effects on substrate localization and activity of both integral membrane and soluble proteins. This study showed that the palmitoylation level was decreased in the fibrotic kidneys of mice and patients with CKD. The expression of DHHC9 was down-regulated, whereas that of APT1 was upregulated in TECs of CKD. Adenovirus-mediated overexpression of DHHC9 or ipronjazid treat-ment and the deletion of APT1 in TECs or inhibition of APT1 using ML348 treatment effectively alleviated UUO- or IRI-induced renal fibrosis. The deletion of DHHC9 in the tubule resulting in the down-regulation of enzymatic palmitoylation in TECs, could significantly aggravate kidney fibrosis (Fig. 8). In contrast to the previously pro-posed phosphorylation[20], and the passive and non-enzymatic meta-bolite PTMs such as acetylation[21], malonylation[22] and succinylation[23], this study provided simple, specific, and easily druggable neo-targets for the treatment of patients with CKD.

Previous studies have demonstrated that TGF-β increases β-catenin activity by inhibiting GSK3β activity[24], promoting interac-tions between Smad2/3 and β-catenin[25], phosphorylating β-catenin at Y654[26], upregulating Wnt/β-catenin signaling through effects on the ligands, dickkopff-1, and receptor complexes[27]. This study demon-strated that TGFβ downregulates DHHC9 and upregulates APT1, which results in reduced β-catenin palmitoylation and the activation of β-catenin to promote renal fibrosis.

This study, for the first time, reported the palmitoylation of β-catenin at Cys 300 and demonstrated that DHHC9 and APT1-regulated palmitoylation of β-catenin at Cys300 affected its stability and nuclear translocation (Fig. 8). In the absence of canonical Wnt ligands, β-catenin in the cytosol is phosphorylated and degraded via the proteasomal pathway[16]. The deactivation of the degradation complex leads to the accumulation of β-catenin, which undergoes nuclear translocation to initiate the transcription of downstream genes[28]. This highlights the effect of palmitoylation on β-catenin degradation regardless of Wnt activation and further improved our understanding of the β-catenin degradation pathway. Although this study demonstrated that the palmitoylation of β-catenin Cys300 promotes its degradation, the underlying mechanisms were not elu-cidated. The mechanism may involve binding to the proteins in the degradation complex, or the inclusion of a hydrophobic tag in the protein that promotes protein degradation.

It is well known that β-catenin accumulates in the cytoplasm and subsequently enters the nucleus. However, the mechanism through which β-catenin enters the nucleus remains unclear. β-catenin, which does not contain nuclear localization signal or nuclear export signal (NES) sequences, appears to be imported into the nucleus in an importin/karyopherin-independent manner, potentially by directly interacting with the nuclear pore components[29]. Conversely, APC and AXIN1 harbor functional NES sequences and facilitate the nuclear export of β-catenin[30,31]. The intracellular distribution of β-catenin may also be influenced by its cytoplasmic and nuclear retention via inter-actions with the binding partners[32]. Therefore, the role of palmitoyla-tion in the nuclear entry or exit and degradation of β-catenin is unclear.

Moreover, the results of the APE assay suggested that β-catenin has multiple S-acylation sites. β-catenin palmitoylation was markedly downregulated when Cys 300 was mutated or when Cys 520 or Cys 573 was mutated. The results of the TCF/LEF luciferase reporter assay implied that the palmitoylation of the β-catenin at Cys 520 site enhanced its nuclear translocation and upregulated TCF/LEF1-depen-dent gene expression. If palmitoylation occurs at Cys 520, the reg-ulation of β-catenin via palmitoylation will be complex. Further experiments are needed to verify this. Additionally, the presence of another palmitoyltransferase must be explored in the future.

In the early 1950s, iproniazid was introduced for the treatment of tuberculosis[33]. However, it was withdrawn from the United States market in 1961 after being reported to induce hepatitis and jaundice[34]. In this study, drugs that exerted effects in contrast to those of *Zdhhc9* knockout on cell survival were screened across the Cancer Depen-dency Map. Iproniazid was selected as it exerted effects similar to DHHC9 on cell survival. In vitro studies demonstrated that 200 nM iproniazid treatment markedly upregulated DHHC9, downregulated β-catenin, and inhibited TGF-β1-induced FN production in TECs. Animal studies demonstrated that iproniazid mitigated UUO- or IRI-induced renal fibrosis at a dose of 0.3 mg/kg bodyweight/d, which is much lower than the standard dosage used in clinical practice. Therefore, low-dose iproniazid may be a promising drug for CKD.

This study reported that β-catenin deletion in tubular cells alle-viated UUO or IRI-induced renal fibrosis. Dong Zhou et al. have reported that the deletion of β-catenin did not alter the severity of renal fibrosis after obstructive injury[35]. However, Dong et al. sacrificed mice on day 3 or 7 post-operation. While mice were sacrificed on day 10 post-operation in this study. Stellor Nlandu-Khodo et al. reported that mice

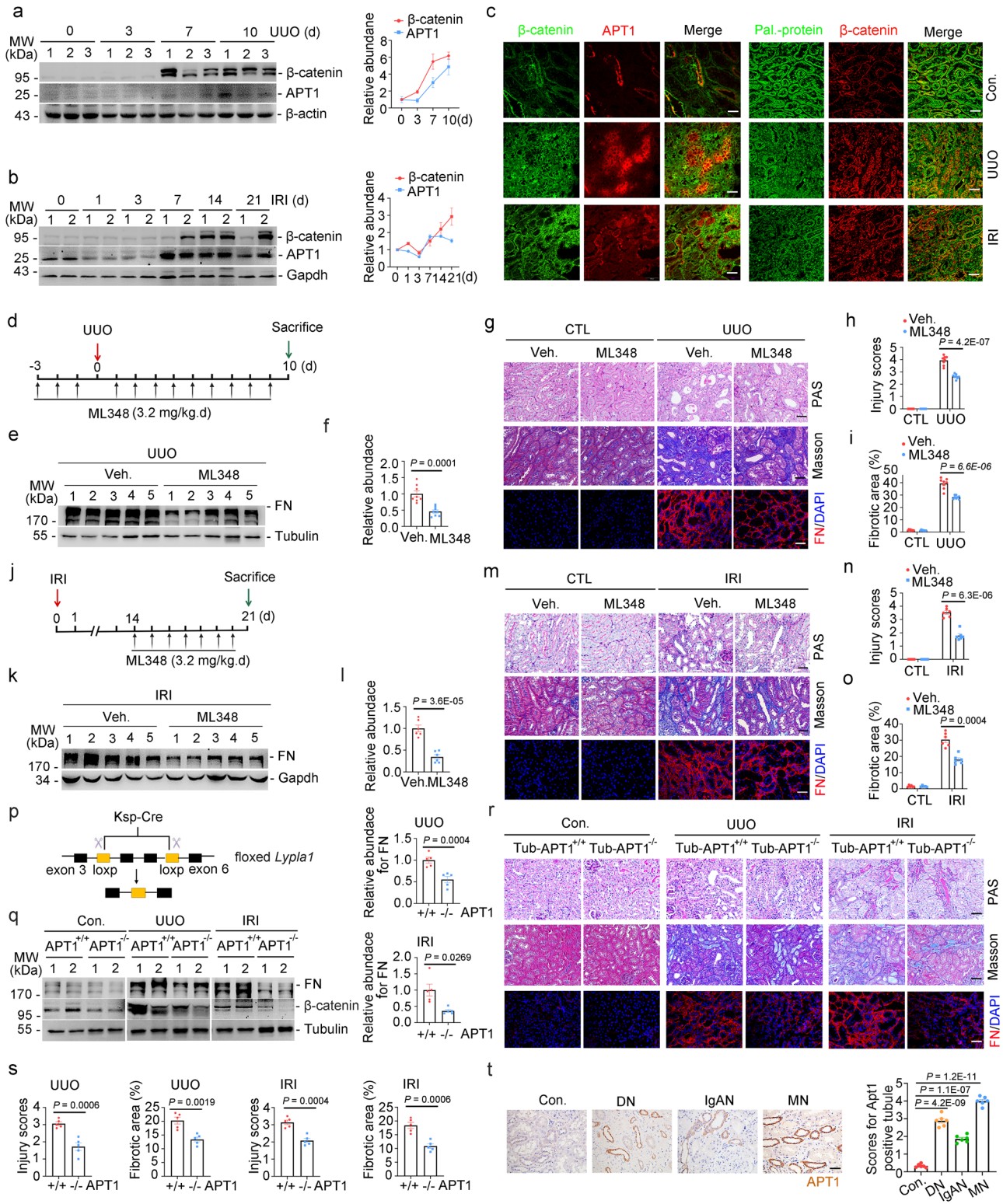

with increased β-catenin activity specifically in the proximal tubule were protected from kidney fibrosis and epithelial injury[36]. Nlandu-Khodo et al. generated mice with increased β-catenin activity in which the phosphorylation site targeting β-catenin for degradation is floxed. In addition, animal models they utilized were aristolochic acid (AA), or unilateral IRI followed by contralateral nephrectomy, which are different from the models we used. Moreover, Nlandu-Khodo et al. generated mice with increased β-catenin activity specifically in the proximal tubule. In contrast, we generated mice with β-catenin ablation mainly in

the distal tubule as well as in partial proximal tubule. Previous studies have linked ongoing epithelial cell death to CKD progression in injured kidneys[37]. And β-catenin signaling may promote epithelial survival in AKI[38]. Therefore, it is possible that increased β-catenin activity in the proximal tubule may protect against acute kidney injury at the first stage, and subsequently alleviate the progression of CKD.

This study reported that DHHC9 was downregulated and APT1 was upregulated in the renal tubules of patients with CKD. It's essential to acknowledge that due to the knockout efficiency of target genes in the

**Fig. 7 | APT1 induction accelerates kidney fibrosis in mice.** Western blot and quantitative analyses for APT1 and β-catenin in kidneys after UUO (**a**) or IRI (**b**). *n* = 3 biologically independent animals. **c.** Representative immunofluorescent images showing the co-localization of APT1 and β-catenin in mouse kidneys after UUO or IRI. Scale bar, 20 μm (left). Representative images for protein palmitoylation and co-staining of β-catenin in mouse kidneys. Scale bar, 20 μm (right). **d** Strategy for UUO surgery and ML348 administration. **e, f** Western blot assay and quantitative analysis for FN in UUO kidneys. *n* = 8. Representative images of PAS, Masson-trichrome and FN staining. Scale bar, 20 μm (**g**), injury scores (**h**) and the fibrotic area (**i**) in kidneys after UUO. *n* = 8. **j** Strategy for IRI surgery and ML348 administration. **k, l** Western blot assay and quantitative analysis for FN in IRI kidneys. *n* = 6. Representative images of PAS, Masson-trichrome and FN staining. Scale bar, 20 μm (**m**), injury

scores (**n**) and the fibrotic area (**o**) in kidneys after IRI. *n* = 6. **p** Strategy for generating mice with tubule APT1 deletion. **q** Western blot analyses for FN and β-catenin and quantitative analyses for FN in kidneys as indicated. *n* = 5.
**r** Representative images of PAS, Masson-trichrome and FN staining in kidneys as indicated. Scale bar, 20 μm. **s** The graph showing the injury scores and the fibrotic area among groups. Each dot represents the average of five HPFs from each mouse. *n* = 5. **t** Representative immunohistochemical staining images and quantitative analyses for APT1 in tubule from CKD patients including diagnosis of Diabetic nephropathy (DN), IgA nephropathy (IgAN) and Membranous nephropathy (MN). Scale bar, 20 μm. *n* = 6. Representative results were obtained from at least three independent experiments with similar results. Data represent the mean ± SEM. *P* values were determined by the two-tailed Student's *t*-test.

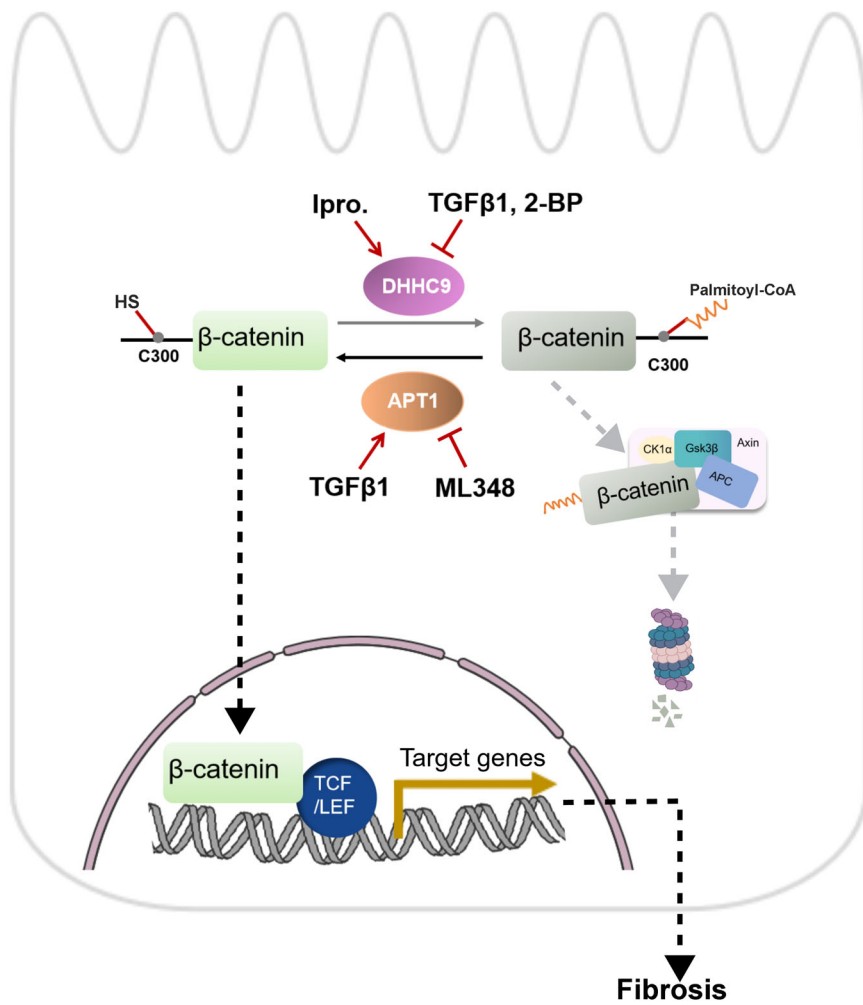

**Fig. 8 | The schematic diagram of DHHC9 and APT1-mediated β-catenin palmitoylation at Cys 300 affecting its degradation and kidney fibrosis.** DHHC9, Asp-His-His-Cys (DHHC) motif-containing palmitoyl *S*-acyltransferase 9; APT1, acyl protein thioesterase 1; Ipro., iproniazid; TGFβ1, Transforming growth factor β1; 2-

BP, 2-bromopalmitate; CK1α, casein kinase 1α; GSK3β, glycogen synthase kinase 3β; APC, adenomatous polpyposis coli; TCF/LEF, T cell factor/lymphoid enhancer factor.

distal tubules significantly surpasses that in the proximal tubules through the use of Ksp cre transgenic mice, the role of DHHC9 in the regulation of kidney fibrosis might potentially be underestimated, as opposed to overestimated, within the scope of this study using these transgenic mice. The kidneys were enriched with PATs (Fig. 2a). In this study, we focused on DHHC9. However, the other PATs, such as *ZDHHC6* (preferentially expressed in podocytes) (Fig. 2d) and *ZDHHC1, 2, 5* (up-regulated in the fibrotic kidneys) (Fig. 2b, Supplementary Fig. 2 and Fig. 3), may also contribute to renal fibrosis by regulating palmitoylation of different substrates in various cell types. Furthermore,

other than β-catenin, DHHC9 may also modulate other substrates that are involved in renal fibrosis. RAS proteins have been reported to be the substrates of DHHC9[39,40]. DHHC9 also palmitoylates Glut1 and facilitates its membrane localization and glucose uptake[41]. Moreover, DHHC9 may have other substrates such as IL-20R, CREB, IL-4, and ARG1, which may be related to fibrosis (Supplementary Fig. 13).

In summary, the findings of this study demonstrated that DHHC9 and APT1 are promising therapeutic targets for CKD. Therefore, specific small molecules that activate DHHC9 or inactivate APT1 should be developed in the future.

# Methods

## Study approval

All animals were maintained in the Specific Pathogen-Free Laboratory Animal Center of Nanjing Medical University, according to the guidelines of the Institutional Animal Care and Use Committee from Nanjing Medical University (IACUC-2112054).

## Animals

The Ksp-Cre transgenic mice were ordered from Jackson Laboratory (cat: 012237; C57BL/6J background). Mice with a floxed DHHC9 allele (DHHC9 floxed mice; exon 3 flanked by loxP sites) and mice with a floxed APT1 allele (APT1 floxed mice; exon 4 to 5 flanked by loxP sites) were ordered from Cyagen (cat: S-CKO-05388 and S-CKO-04319, respectively; C57BL/6 J background). Mice with a floxed β-catenin allele (β-catenin floxed mice; exon 8 to 13 flanked by loxP sites) were ordered from Jackson Laboratory (cat:022775; C57BL/6 J background). Zdhhc9, APT1, or β-catenin floxed mice were crossed with Ksp-Cre mice respectively to generate offsprings with tubular cell deletion of DHHC9 (Tub-DHHC9$^{-/-}$, genotype: Cre$^{+/-}$, DHHC9$^{fl/fl}$), APT1 (Tub-APT1$^{-/-}$, genotype: Cre$^{+/-}$, APT1$^{fl/fl}$), or β-catenin (Tub-β-catenin$^{-/-}$, genotype: Cre$^{+/-}$, β-catenin$^{fl/fl}$), respectively. The same gender with genotyping Cre$^{-/-}$, from the same litters were considered as control littermates. Taking DHHC9 deletion model as an example, Tub-DHHC9$^{-/-}$ and control littermates, aged between 8 and 10 weeks, were subjected to UUO or IRI operation to induce kidney fibrosis. Mice were housed in a pathogen-free environment with the temperature maintained at $23 \pm 2\,^{\circ}C$ and relative humidity at 50 to 65% under a 12 h/12 h light/dark cycle with free access to food and water. Mice were maintained on normal chow (cat:19123123, Beijing Keao Xieli Feed Co., Beijing, China).

## UUO and IRI surgery

Male C57BL/6 mice weighing approximately 18–20 g were acquired from Vital River Laboratory. Adenovirus virus (Ad) constructs of Zdhhc9 (Ad-DHHC9-HA) and negative control (Ad-vector) were purchased from HanBio (Shanghai, China). Adeno-associated virus (AAV) constructs containing β-catenin WT or C300S mutant were purchased from EKBIO (Nanjing, China). Ad-DHHC9-HA was delivered to kidneys by intra-parenchymal injection according to the previous study[42]. Briefly, in anesthetized mice, a needle (gauge 31) was inserted in the lower pole of the kidney parallel to the long axis and was carefully pushed toward the upper pole. As the needle was being slowly removed, 50 μl of filter-purified adenovirus was injected. The intra-parenchymal injection was conducted at day 3 after UUO surgery or day 10 after IRI surgery. UUO was performed using an established procedure[43]. For kidney IRI surgery, the left renal pedicle of the mouse was clamped for 30 minutes and then released. The right kidney was not removed. Mice were euthanized by cervical dislocation, and kidneys were harvested at different time points after surgery as indicated.

## Cell culture and transfection

For preparing the primary cultural tubular cells (PTCs), mouse kidney tissues were filtered through the 150-μm mesh and digested with 2 mg/ml collagenase I for 30 min at 37ºC with gentle stirring. Cells were harvested and cultured in Dulbecco's modified Eagle's medium-F12 (DMEM-F12) (cat: 12400024, Gibco, Grand Island, NY) supplemented with 10% FBS (cat: FND500, ExCell Bio, Shanghai, China), 0.5 x Insulin-Transferrin-Selenium (Glibco, Grand Island, NY), 36 ng/ml Hydrocortisone, 4 pg/ml Triiodothyronine, 10 ng/ml Epidermal Growth Factors (Sigma-Aldrich, St. Louis, MO), and 1% penicillin-streptomycin at 5% $CO_2$, 37 ºC to 80% confluence, and were digested with trypsin and planted. The primary cultured tubular cells with floxed target gene were infected with Ad-Cre to ablate target gene. Small interfering RNAs specific for mouse Zdhhc9 and APT1, respectively, were ordered from Shanghai Integrated Biotech Solutions Co., Ltd. pCMV6-DDK-tagged β-catenin WT and its mutant plasmids, pCMV6-DDK-APT1, pCMV6-

EGFP-β-catenin WT and its C300S mutant (Miaolingbio, Wuhan, China) were transfected into PTCs or HEK293A cells using Lipofectamine 3000 reagent (Invitrogen, Grand Island, NY). Cells were harvested and analyzed at 36 h after transfection.

## Reagents and antibodies

The following reagents and antibodies were employed: 2-bromopalmitate (2-BP, cat: 21604, Sigma-Aldrich, St Louis, MO), ML348 (cat: HY-100736, MCE, Shanghai, China), Iproniazid (cat: HY-B0886A, MCE), TGF-β1 (cat: 240-B-010-CF, R&D Systems, Minneapolis, MN), MG132 (cat: M8699, Sigma-Aldrich), Lactacystin (cat: sc3575, Santa Cruz), ICG001 (cat: A8217, Apexbio, USA), Recombinant Human DHHC9 protein (cat: ab162420, abcam), Recombinant Human β-catenin protein (cat: 63175, abcam), Anti-fibronectin antibody produced in rabbit (1:5000; cat: F3648, Sigma-Aldrich;), Mouse monoclonal anti-Tubulin (1:5000; RRID: AB_630403, cat: sc53646, Santa Cruz), Anti-DHHC9 antibody produced in rabbit (1:500; cat: SAB4502104, Sigma-Aldrich), HA-Tag (6E2) Mouse mAb (1:1000; cat: 2367S, Cell signaling technology), GAPDH polyclonal antibody (1:3000; cat: AP0063, Bioworld, Nanjing, China), β-Catenin (6B3) Rabbit mAb (1:3000; cat: 9582S, Cell signaling technology), Phospho-β-Catenin (Ser33/37) Antibody (1:1000; cat: 2009S, Cell signaling technology), Non-phospho (Active) β-Catenin (Ser45) (D2U8Y) Rabbit mAb (1:1000; cat: 19807S, Cell signaling technology), Anti-Cyclin D1 (1:1000; cat: 2922S, Cell signaling technology), Monoclonal ANTI-FLAG® M2 antibody produced in mouse (1:3000; cat: F1804, Sigma-Aldrich), Na, K-ATPase α1 (D4Y7E) Rabbit mAb (1:1000; cat: 23565S, Cell signaling technology), Histone H3 (D1H2) XP® Rabbit mAb (1:1000; cat: 4499S, Cell signaling technology), Axin1 (C76H11) Rabbit mAb (1:500; cat: 2087S, Cell signaling technology), Casein kinase I alpha antibody (H-7) (1:1000; cat: sc-74582, Santa Cruz Biotechnology), GSK-3β (27C10) Rabbit mAb (1:1000; cat: 9315S, Cell signaling technology), Anti-Ubiquitin (1:500; cat: sc-8017, Santa Cruz), LYPLA1 (K90) polyclonal antibody (1:500; cat: BS3063, Bioworld, Nanjing, China).

## In vitro protein acyltransferase (PAT) assay

Palmitoyl acyltransferase constituted by recombinant DHHC9 (cat: ab162420, abcam) (100 ng) were incubated with β-catenin (cat: ab63175, abcam) (2 μg) in 25 μl of reaction buffer (50 mM Tris-HCl [pH 7.4], 10 μM palmitoyl azide-CoA, 1 μM palmostatin B) at 25 °C for 1 h followed by a Cu(I)-assisted click reaction with biotin picolyl alkyne (50 μM) to biotinylate the proteins with the incorporation of palmitoyl azide. The samples were loaded onto 30 kDa spin column (cat: FUF053-5pcs, Beyotime Biotechnology, Shanghai, China) to remove the free biotin picolyl azide using a buffer containing 50 mM Tris-HCl [pH 7.4], 150 mM NaCl, 0.5 mM EDTA, 0.1% Triton X-100, 0.1% SDS, and 0.5% NP40. Biotinylated proteins were captured by streptavidin agarose beads prior to boiling in SDS-PAGE sample buffer without DTT for 10 min at 95 °C. Immunoblotting was performed to analyze the palmitoylation of the target proteins.

## Click-iT identification of β-catenin palmitoylation

At 36 h after β-catenin WT or mutant plasmid transfection, 100 μM of Click-iT palmitic acid and azide (cat: C10265, Thermo Fisher Scientific) were added to the cell medium with gentle mixing, then incubated at 37 °C and under 5% CO2 for 6 h. Then, the medium was removed, and the cells were washed three times with 1×PBS before the addition of lysis buffer (1% sodium dodecyl sulfate in 50 mM Tris-HCl, pH8.0) containing protease and phosphatase inhibitors at appropriate concentrations. Cells were incubated on ice for 20 min, then tilted the plates and pipetted the lysate into a 1.5 ml microcentrifuge tube. Then, the lysates were sonicated with a probe sonicator to solubilize the proteins and disperse the DNA. After vortexing the lysate for 5 mins and centrifuging the cell lysate at 13,000-18,000 g at 4 °C for 5 mins, the supernatant were transferred to a clean tube and the protein

concentration was determined using the BCA Protein Assay Kit (cat: 23225, Thermo Fisher Scientific). Thus, the protein sample was reacted with biotin-alkyne using the Click-iT Protein Reaction Buffer Kit (cat: C10276; Thermo Fisher Scientific) following the instruction. Then, biotin alkyne-azide-plamitic-protein complex was pulled down by streptavidin and, after washing, the pellets were subjected to immunoblotting for β-catenin.

## Acyl−biotin exchange

Acyl−biotin exchange (ABE) assays were performed essentially according to the procedure[44]. Samples were suspended in 1 ml lysis buffer (50 mM Tris-HCl pH 7.4, 5 mM EDTA, 150 mM NaCl, 2.5% SDS, inhibitor cocktail). Then, the lysates were sonicated with a probe sonicator, and the protein concentration of the supernatant was determined using BCA Protein Assay Kit. Protein (200 μg) for each sample treated with 10 mM of TCEP (cat: T2556, Thermo Fisher Scientific) 30 minutes, followed by 25 mM N-ethylmaleimide (NEM) (cat: 23030, Thermo Fisher Scientific) addition. Samples were vortexed at RT for 2 h and precipitated with chloroform/methanol/water (v/v/v 1:4:3), briefly air-dried, and dissolved in 1 ml of lysis buffer with 5 mM biotin-HPDP (cat: 21341, Thermo Fisher Scientific) by gently mixing at RT. Samples were then equally divided into two parts and incubated with 0.5 ml of 1 M hydroxylamine (HA) or negative control (1 M NaCl) respectively at RT for 3 h. Samples were precipitated again and dissolved in 200 μl of resuspension buffer (50 mM Tris-HCl pH 7.4, 2% SDS, 8 M urea, 5 mM EDTA). For each sample, 20 μl was used as loading control and 180 μl was diluted 1:10 with 1×PBS and incubated with 20 μl of streptavidin beads (cat: 20357, Thermo Fisher Scientific) with shaking overnight at 4 °C. Beads were washed 3 times with 1×PBS containing 1% SDS. The beads and loading controls were mixed with SDS loading buffer and heated at 95 °C for 10 min. Samples were then resolved by SDS-PAGE and subjected to western blot analyses.

## Histological Analysis of Renal Tissues

Patient renal biopsies and the mouse kidneys were fixed with 4% paraformaldehyde (PFA) at 4 °C overnight. Tissues were embedded in paraffin and cross-sectioned (3-μm) for histology examination. Immunohistochemistry analyses were performed as previous study[45]. Ten randomly selected high-powered fields were viewed per specimen, with each field scored on a 5-point scale (0, weakest; 5, strongest) based on the staining intensity. The average score of each specimen was calculated. Periodic-acid-Schiff (PAS), Masson, Sirius Red staining were performed according to manufactures' instructions. The sections were stained with appropriate primary antibodies. The samples of renal biopsies were obtained from Center for Kidney Diseases, the Second Affiliated Hospital of Nanjing Medical University. CKD diagnoses included Diabetic nephropathy (DN), IgA nephropathy (IgAN) and Membranous nephropathy (MN). The use of patient specimens and written informed consent of the human samples were approved by the Institutional Review Board at the Second Affiliated Hospital of Nanjing Medical University.

## Acyl-PEG exchange assay

Acyl-PEG exchange assay (APE) was performed according to the procedure with slight modifications[46]. In brief, trypsinized cells were washed three times with 1×PBS and lysed in lysis buffer (5 mM triethanolamine, 150 mM NaCl, 4% SDS, 2 units benzonase, 5 mM PMSF, pH 7.4) with protease inhibitor cocktail followed by adding EDTA to a final concentration of 5 mM. The cell lysates were treated with 10 mM of TCEP for 30 min and then incubated with 25 mM of NEM for 2 h at RT to reduce disulfide bonds and cap the free cysteine residues, respectively. The mixture was precipitated by the sequential addition of methanol, chloroform, and distilled water (v/v/v, 4/1.5/3) into the 1.5-ml Eppendorf tube. The protein pellets were washed twice with prechilled methanol, resuspended in TEA buffer (5 mM Triethanolamine, 150 mM NaCl, 4% SDS, 4 mM EDTA, pH 7.4), and then incubated with 0.75 M of $NH_2OH$ for 1 h at RT to cleave palmitoylation thioester bonds. $NH_2OH$ was removed by methanol-chloroform-water precipitation and the protein pellets resuspended in TEA buffer with 0.2% Triton X-100 were incubated with 1 mM of mPEG-Mal (cat: JKA3115, Sigma-Aldrich) for 2 h at RT. The samples were precipitated again by methanol-chloroform-water and then resuspended with 2×SDS without DTT, boiled at 95 °C for 3 min, separated by SDS-PAGE, and analyzed by immunoblotting.

## Subcellular fractionation

Cell fractions were obtained from the tubular cells using the cell fractionation kit (cat: 9038, Cell signaling technology). Briefly, cells were washed with 1×PBS and trypsinized. Cold medium was added, and cells were centrifuged for 5 minutes at 350 g. Cells were washed with cold 1×PBS, resuspended in 500 μl 1×PBS and centrifuged at 4 °C for 5 mins at 500 g. The pellets were resuspended in 300 μL ClB buffer, incubated on ice for 5 minutes and centrifuged at 4 °C for 5 min at 500 g. The supernatant represents the cytoplasmic fraction. Pellets were then resuspended in 300 μL MlB buffer, incubated 5 minutes on ice and centrifuged at 4 °C for 5 minutes at 8000 g. The supernatant represents the membrane and organelle fraction. Pellets were then lysed with buffer (1% NP40, 50 mM Tris pH 7.5, 300 mM NaCl, 150 mM KCl, 5 mM EDTA, 1 mM DTT, 10 mM PMSF, 10% glycerol) for 30 minutes on ice. Cells were centrifuged at 4 °C for 20 minutes at 20,000 g. The supernatant constitutes the nuclear fraction. Equivalent portions of different fractions were subjected to western blot analyses.

## Immunoprecipitation and immunoblotting

PTCs or HEK293A cells were washed with cold 1×PBS for three times and lysed with IP lysis buffer (Beyotime, Shanghai, China) containing 1% protease inhibitor cocktail and 1% phosphatase I and II inhibitor cocktail (Sigma Aldrich, St Louis, USA) on ice for 10 minutes. The supernatants were collected after centrifugation at 16000 g at 4 °C for 15 minutes. Protein concentration was determined by BCA Protein Assay Kit (Pierce Thermo-Scientific, Rockford, IL). An equal amount of protein (500 · g) was incubated overnight at 4 °C with 1 ug primary antibody. After incubating with A/G PLUS-Agarose beads (Santa Cruz, Dallas, TX) for 2 hours, the beads were washed with 1×PBS and eluted with 2×SDS sample buffer for immunoblotting or LC-MS analysis. For immunoblotting, samples were resolved on 10% or 15% polyacrylamide mini-gels and transferred onto a nitrocellulose filter membrane. The membranes were probed with primary and HRP-conjugated secondary antibodies. Immunoblots were visualized by enhanced chemiluminescence detection (Vazyme Biotech Co., Ltd). Densitometry analyses of immunoblots were performed using Image J software.

## Luciferase activity assay

The effect of DHHC9 or APT1 on β-catenin−mediated transcription was detected by using the TOP/FOP-flash TCF reporter plasmid containing two sets of three copies of the TCF binding site upstream of the thymidine kinase minimal promoter and luciferase open reading frame (Millipore). PTCs were seeded on 12-well culture plates to 80%−90% confluence in complete medium containing 10% FBS and changed to serum-free medium after washing twice with serum-free medium. PTCs were transfected with DHHC9 or APT1 siRNA and TOP-flash or FOP-flash plasmid (1 · g) and an internal control reporter plasmid (0.1 g) Renilla reniformis luciferase driven under thymidine kinase promoter (pRL-TK, Promega) using Lipofectamine 3000 (Invitrogen) according to the manufacturer's instruction. Twenty-four hours later, cells were treated with TGFβ1 (2 ng/ml) for 12 hours. Luciferase assay was performed using a dual luciferase assay system kit according to the manufacturer's protocol (Promega). Relative luciferase activity was reported as fold induction over the controls.

### Real-time qRT-PCR assay

Total RNA was extracted using TRIzol Reagent (Invitrogen, USA) according to the manufacturer's instruction. cDNA was synthesized using 1 µg of total RNA, ReverTra Ace (Vazyme, Nanjing, China) and oligo (dT)12–18 primers according to the manufacturer's protocol. Gene expression level was measured by real-time qRT-PCR (Vazyme, Nanjing) and Roche real-time PCR system (LightCycler 96). The relative amount of mRNA to internal control was calculated using the equation $2^{\Delta CT}$, in which $\Delta CT = CT_{gene} - CT_{control}$.

### Immunofluorescence staining and confocal microscopy

Cells were seeded in 35-mm glass bottom dishes (MatTek) and fixed with 4% paraformaldehyde (v/v in PBS) for 30 mins. The fixed cells were washed twice with 1×PBS, permeabilized and blocked with 4% BSA for 45 mins. The permeabilized cells were incubated overnight at 4 °C with primary antibody, followed by incubation with secondary antibody at RT in the dark for 1 h. Kidney cryosections were fixed with 3.7% paraformaldehyde for 15 mins at RT and immersed in 0.2% Triton X-100 for 10 mins. After blocking with 10% donkey serum in 1×PBS for 1 h, slides were immunostained with primary antibodies. Samples were stained with DAPI (cat: C0065S, olarbio) and observed.

### Click-iT fluorescence identifying protein palmitoylation in kidneys

To visualize renal palmitoylation, we injected mice with Click-iT palmitic acid-azide intraperitoneally on day 9 after UUO or on day 20 after IRI. We typically administered Azide-labeled palmitic acid intraperitoneally at 8 a.m. on the day prior to the experimental endpoint, followed by subsequent injections at 6-hour intervals. At 6 hours after the fourth injection, mice were euthanized, and kidney tissues were collected for subsequent analyses. Kidney cryosections were performed the click reaction with an azide or alkyne and the corresponding click detection reagent according to The Click-iT® Cell Reaction Buffer Kit (cat: C10269, Thermo Fisher Scientific) protocol. Briefly, fixed and permeabilized cells following a standard protocol, washed cells once with 1–3% BSA in 1×PBS, prepared the Click-iT® reaction cocktail within 15 minutes, added 0.5 mL Click-iT® reaction cocktail to each sample, incubated samples at RT for 30 minutes, protected from light, washed cells once with 1-3% BSA in 1×PBS, stained with a desired counter stain or antibodies prior to imaging.

### GST pull-down assay

His-tagged purified proteins (200 ng) were mixed with GST fusion proteins (100 ng) or GST protein as a control in the binding buffer (50 mM Tris-HCl [pH 7.5], 1% Triton X-100, 150 mM NaCl, 1 mM DTT, 0.5 mM EDTA, 100 µM PMSF, 100 µM leupeptin) for 1 h and incubated with glutathione agarose beads (cat: 16100, Thermo Fisher Scientific) for additional 30 min at 4 °C. The bound protein complexes were retained by washing the beads with binding buffer three times and detected by immunoblots.

### Statistical analysis

Statistical analyses were performed using GraphPad Prism 9.0.2 Software. All values are expressed as means and SEM. Paired or unpaired $t$-test was used to compare the two groups. The sample size in each study was based on previous studies. A $P$ value of 0.05 or lower was considered statistically significant.

### Reporting summary

Further information on research design is available in the Nature Portfolio Reporting Summary linked to this article.

## Data availability

The authors declare that all data supporting the findings of this study are available within the article and its Supplementary Information files, or from the corresponding author. Databases used in this study include GEO Datasets (GSE12501, https://www.ncbi.nlm.nih.gov/geo/query/acc.cgi?acc=GSE125015; GSE98622; GSE69438), Human Protein Atlas (https:// www.proteinatlas.org), Database of Protein, Genetic and Chemical Interactions (https://thebiogrid.org/), Database on Protein S-palmitoylation (https://swisspalm.org/) and Kidney Single Cell Datasets (http://humphreyslab.com/SingleCell/). Source data are provided with this paper.

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

## Acknowledgements

This work was supported by National Science Foundation of China Grants 81970627/H0504; 82370688/H0502; Jiangsu Province social development fund BE2020725; "333" Project; Six talents summit program WSN-065 to Dai C.

## Author contributions

M.G. performed experimental procedures, conception and construction of the study, manuscript writing, and data analysis; H.J., L.Y., N.X., M.T. participated in in vivo and in vitro experiments; Y.L., H.W., Q.H. participated in animal model construction; C.D. supervised the entire project, designed the experiment, analyzed the data, revised the manuscript, and approved the final version of the manuscript for publication.

## Competing interests

The authors declare no competing interests.
