## [Peer Review File · Nature Communications]

Palmitoyltransferase DHHC9 and acyl protein thioesterase 1
modulate renal fibrosis through regulating β -catenin
palmitoylationREVIEWER COMMENTS

Reviewer #1 (Remarks to the Author):

This is a very interesting study demonstrating the role and mechanism of protein palmitoylation in regulating beta-catenin signaling and kidney fibrosis. The authors found that both protein palmitoylation and DHHC9 palmitoyltransferase were downregulated in mouse models of renal fibrosis and patients with chronic kidney diseases (CKD). Utilizing both gain- and loss-of function approaches, they further demonstrated that protein palmitoylation generally mitigated kidney fibrosis. They went on showing that beta-catenin was the direct substrate of protein palmitoylation, and palmitoylated beta-catenin was tagged for ubiquitination and degradation. They concluded that protein palmitoylation plays a role in regulating kidney fibrosis.

Overall, this study is novel, comprehensive and well presented. It describes an important role of protein palmitoylation in renal fibrogenesis and elucidates its potential target and mechanism. The studies are comprehensive and investigated both palmitoyltransferase DHHC9 and de-palmitoylation enzyme APT1, using gain- and loss-of function approaches, combined with genetic and pharmacological inhibition. The conclusion is supported by abundant experimental evidence. There are some questions needing to be addressed.

Specific comments:

1. It is well known that Wnt ligands are lipid-modified proteins, presumably subjected to palmitoylation as well. What is the status of Wnts palmitoylation in CKD? Does the palmitoylation of Wnt ligands is also altered in the fibrotic kidney? This is an interesting and important question, as this study focused on beta-catenin signaling.
2. Figure 2F. The quality of this Western blot is very poor. There were several bands with similar intensity and different sizes in the images. How can the authors be confident that the band indicated is the DHHC9 protein as claimed? Similarly, the immunostaining for DHHC9 (Figure 2G and I) is also not convincing. It is difficult to identify positive cells in the control kidney, where is supposedly to be with high level of expression.

3. DHHC9 presumably can cause palmitoylation on many protein substrates. While it is reasonable for focusing on beta-catenin in the present study, authors need to discuss other potential substrates.

4. The role of tubular beta-catenin in kidney fibrosis is controversial, with inconsistent reports showing that tubule-specific deletion of beta-catenin causes detrimental, beneficial and no effect on the severity of renal fibrosis after injury. This raises a question whether altered palmitoylation on other proteins playing a role in this process.

Reviewer #2 (Remarks to the Author):

The author present a thoughtful and thorough analysis of a novel mechanism that regulates beta catenin expression and fibrosis in the setting of kidney injury (UUO and IRI). The finding palmitoylation has a role in regulating a catenin is not novel. However, the fact that a specific palmitoyltransferase, DHHC9, palmitoylates beta catenin and regulates its expression by promoting proteasomal degradation is novel, as is the regulation of beta catenin in the setting of kidney injury by palmitoylation (via DHHC9) and the release of palmitate by an acyl protein thioesterase. This is an impressive body of work. I have a number questions and concerns that the authors should address.

What was the rational for using the Ksp Cadherin cre mouse to generate tubular specific APT1, DHHC 9 and beta catenin knockouts? As the authors noted, this will result in a knockout in more distal segments of the nephron. Your DHHC 9 expression data in Fig 3 suggest that it is primarily expressed in prox tubular segments under baseline conditions and early after IRI injury. To me, this is a concern that the authors should carefully address. For tubular KO's, why was a Pax8 cre or a proximal tubule expressed Cre not used? In which tubular segments is beta catenin palmitoylation occurring that is affecting fibrosis in the injury models?

Would a mouse bearing a beta catenin C300S mutation exhibit enhanced kidney injury, assuming the mice are viable? This would be a key reagent to test the authors central

hypothesis.

Quantification of key experiments in Figs 5-7, and in supplemental figures should be provided, as you are only proving data from representative experiments. For example, I would like to see quantification of the data presented in Fig 5H, where you present data suggesting that C300 on beta catenin is a key palmitoylation site. There should be reduced palmitoylation of beta catenin in cells treated with a DHHC9 siRNA (5E). I am not appreciating this when normalized to a loading control, although it may be reduced when normalized to total beta catenin in the representative figure. I am also not seeing changes in beta catenin levels in cells transfected with an APT1siRNA (6G). In 6N, I do not appreciate differences in beta catenin expression across the conditions. Data for Fig 7Q and 7R should be quantified.

Which is more relevant in examining the effects of DHHC9 on fibrosis, total or phosphorylated beta catenin?

I am having difficulty reviewing your figures based on information provide in the figure legend.

While your data provide strong evidence for a role of DHHC9 in palmitoylating beta catenin and targeting it for degradation, I would be cautious in ascribing all the affects you are seeing to DHHC9. DHHCs can assemble as heterooligomers, and it is possible that DHHC9 assembles with other DHHCs in targeting beta catenin for palmitoylation. Also, DHHC9 facilitates palmitoylation of multiple proteins and its possible that palmitoylation of other proteins could influence fibrosis in kidney injury models.

For the Click it assay in mice, you mention that mice received an IP injection 9 after UUO or on day 20 after IRI. You also mention - at 6 hours after the fourth injection ... However, you do not mention when the three additional injections were administered

The detailed point-by-point responses to the reviewer's comments are as the following.

Reviewer #1 (Remarks to the Author):

This is a very interesting study demonstrating the role and mechanism of protein palmitoylation in regulating beta-catenin signaling and kidney fibrosis. The authors found that both protein palmitoylation and DHHC9 palmitoyltransferase were downregulated in mouse models of renal fibrosis and patients with chronic kidney diseases (CKD). Utilizing both gain- and loss-of function approaches, they further demonstrated that protein palmitoylation generally mitigated kidney fibrosis. They went on showing that beta-catenin was the direct substrate of protein palmitoylation, and palmitoylated beta-catenin was tagged for ubiquitination and degradation. They concluded that protein palmitoylation plays a role in regulating kidney fibrosis.

Overall, this study is novel, comprehensive and well presented. It describes an important role of protein palmitoylation in renal fibrogenesis and elucidates its potential target and mechanism. The studies are comprehensive and investigated both palmitoyltransferase DHHC9 and de-palmitoylation enzyme APT1, using gain- and loss-of function approaches, combined with genetic and pharmacological inhibition. The conclusion is supported by abundant experimental evidence. There are some questions needing to be addressed.

Response: Thank you for your positive feedback on our study. We are pleased that you find our study interesting and well-presented.

Specific comments:

1. It is well known that Wnt ligands are lipid-modified proteins, presumably subjected to palmitoylation as well. What is the status of Wnts palmitoylation in CKD? Does the palmitoylation of Wnt ligands is also altered in the fibrotic kidney? This is an interesting and important question, as this study focused on beta-catenin signaling.

Response: Thank you for your insightful comments regarding the potential role of Wnt ligand palmitoylation in renal fibrosis. As you mentioned, Wnt proteins are known to be lipid-modified proteins with Wnt3a being well-studied. Previous studies showed that Wnt3a may be modified with two lipid moieties, a thioester-linked palmitate group on a conserved cysteine residue (Cys77), named S-palmitoylation (PMID:12717451) and a monounsaturated palmitoleic acid moiety linked via an ester bond to a conserved serine (Ser209), named O-acylation (PMID:17141155). However, the X-ray crystal structure of *Xenopus* Wnt8 in complex with the extracellular domain of Frizzled8 revealed that Cys55 on XWnt8 (equivalent to Cys77 on Wnt3a) was involved in a disulfide bond (PMID:22653731) and a significant reduction in acylation was not observed in C77A mutant form (PMID:24292069), raising the question whether Cys77 is really palmitoylated in vivo. In addition, Wnt3a protein was not found in the 23 protein S-palmitoylation public database (<https://swisspalm.org/proteins>, Supplementary Fig. 15a). Notably, our data showed that S-palmitoylation of Wnt3a was not detected by ABE assay in cultured tubular cells (Supplementary Fig. 15b). Together, all these data suggest that Cys77 and the corresponding residues thereof cannot serve as an acylation site, which supports the notion that Wnts may not be S-palmitoylated but only be O-acylated.

It is well known that O-acylation of Wnts by attaching palmitoleate (C16:1) to a conserved Ser in Wnt proteins catalyzed by porcupine, is required for Wnts secretion and signaling (PMID: 24798332), which aggravates renal fibrosis in CKD (PMID: 19297557; PMID: 28336721). Babita Madan et al reported that porcupine inhibitor C59 attenuates renal fibrosis in obstructive nephropathy (PMID: 27083283), suggesting that inhibiting Wnts O-acylation may alleviate renal fibrosis. We agree that investigating the role of Wnt O-acylation in renal fibrosis is an interesting and important question, given previous studies showing that increased Wnts secretion aggravates renal fibrosis in CKD and that

porcupine inhibition attenuates renal fibrosis in obstructive nephropathy. We will explore this possibility in our future studies.

2. Figure 2F. The quality of this Western blot is very poor. There were several bands with similar intensity and different sizes in the images. How can the authors be confident that the band indicated is the DHHC9 protein as claimed? Similarly, the immunostaining for DHHC9 (Figure 2G and I) is also not convincing. It is difficult to identify positive cells in the control kidney, where is supposedly to be with high level of expression.

Response: Thank you for your feedback on the quality of the Western blot and immunostaining for DHHC9 in Figure 2F, G, H, and I. To address your concerns, we repeated the Western blot assay using a new antibody and presented the data in Figure 2f and h. Additionally, we reperformed the immunostaining for DHHC9 and included representative images in Figure 2g and i. We thought that the new data may provide the improved evidence for the expression and localization of DHHC9 in the kidney.

3. DHHC9 presumably can cause palmitoylation on many protein substrates. While it is reasonable for focusing on beta-catenin in the present study, authors need to discuss other potential substrates.

Response: We do appreciate your insightful comments regarding the other potential substrates of DHHC9 in addition to beta-catenin. RAS proteins have been reported as the substrates of DHHC9 (PMID: 16000296; PMID: 24127608; PMID: 17519897). DHHC9 catalyzes Ras palmitoylation and regulates its membrane localization and function (PMID: 33816563; PMID: 23356262; PMID: 25158650). Increased RAS protein cell membrane localization enhances TGFbeta-induced extracellular matrix production (PMID: 32049037; PMID: 31915377). In addition, DHHC9 palmitoylates Glut1 and facilitates its membrane localization and glucose uptake (PMID: 34620861). Furthermore, we found many

other possible substrates of DHHC9 such as IL-20R, CREB, IL-4, and ARG1, that may be related to fibrosis by searching the protein interaction data websites (<https://thebiogrid.org/119302>) (Supplementary Fig.13). We have discussed this in the revised manuscript as suggested.

4. The role of tubular beta-catenin in kidney fibrosis is controversial, with inconsistent reports showing that tubule-specific deletion of beta-catenin causes detrimental, beneficial and no effect on the severity of renal fibrosis after injury. This raises a question whether altered palmitoylation on other proteins playing a role in this process.

Response: We do appreciate your suggestion. Regarding the role of tubule beta-catenin in kidney fibrosis, we agree that there are inconsistent reports on its effects. However, in this study, we found that ablation of β -catenin in tubular cells attenuates TGF β 1-induced FN production (Fig. 4o, p), and UO or IRI-induced renal fibrosis (Supplementary Fig. 7). In addition, by in situ injections of adeno-associated virus (AAV) containing β -catenin WT or C300S in mouse kidneys (Supplementary Fig. 10a, g), we found that AAV-C300S injected mice (Supplementary Fig. 10b, h) had more severe renal fibrosis compared to WT β -catenin injected mice after UO or IRI (Supplementary Fig. 10c-f, i-l). We believe that the controversial surrounding the role of tubular beta-catenin in kidney fibrosis may be due to the animal model and the scarification time after model construction. In addition to beta-catenin, we have searched protein interaction databases to identify other possible substrates for DHHC9 related to fibrosis. Among of them, IL-20R, CREB, IL-4, and ARG1 are very interesting (Supplementary Fig.13) and we will work on these molecules in our future research plan.

Reviewer #2 (Remarks to the Author):

The author present a thoughtful and thorough analysis of a novel mechanism that

regulates beta catenin expression and fibrosis in the setting of kidney injury (UUO and IRI). The finding palmitoylation has a role in regulating a catenin is not novel. However, the fact that a specific palmitoyltransferase, DHHC9, palmitoylates beta catenin and regulates its expression by promoting proteasomal degradation is novel, as is the regulation of beta catenin in the setting of kidney injury by palmitoylation (via DHHC9) and the release of palmitate by an acyl protein thioesterase. This is an impressive body of work. I have a number questions and concerns that the authors should address.

Response: We greatly appreciate the reviewer's recognition of the potential significance of our study and their valuable comments. We recognize that the constructive feedback is crucial for improving the quality of our manuscript.

1. What was the rationale for using the Ksp Cadherin cre mouse to generate tubular specific APT1, DHHC 9 and beta catenin knockouts? As the authors noted, this will result in a knockout in more distal segments of the nephron. Your DHHC 9 expression data in Fig 3 suggest that it is primarily expressed in prox tubular segments under baseline conditions and early after IRI injury. To me, this is a concern that the authors should carefully address. For tubular KO's, why was a Pax8 cre or a proximal tubule expressed Cre not used? In which tubular segments is beta catenin palmitoylation occurring that is affecting fibrosis in the injury models?

Response: We appreciate the reviewer for bringing up this important point. As previously reported by Peter Igarashi, the Cre recombinase driven by the cadherin 16 (cdh16 or Ksp-cadherin) promoter is expressed in collecting ducts, Heinz's loops, distal tubule, as well as weakly expressed in the proximal tubule (PMID: 12089378; PMID: 12089379). This transgenic model has been extensively utilized by many other researchers to achieve target gene deletion in tubular cells (PMID: 22622501; PMID: 32404875; PMID: 35483524; PMID: 36450712; PMID: 33998598 PMID: 32127410).

Single-cell sequencing data show that DHHC9 mRNA is primarily expressed in proximal tubule (Figure 2D and 2E). However, DHHC9 protein was detected in both distal and proximal tubules in the kidneys from mice and CKD patients by immune-staining with anti-DHHC9 antibody (Fig. 2g, i, l), or co-staining with proximal and distal tubule markers (Supplementary Fig. 4c). Data from the Human Protein Atlas confirmed the DHHC9 protein expression in the proximal and distal tubules (Supplementary Fig. 14). Therefore, considering the broad expression of DHHC9 in tubule as well as the characteristics of the disease model, we utilized Ksp-Cre transgenic mice in this study to generate mouse model with tubular cell DHHC9 deletion.

Regarding the use of Pax8 Cre or the other models such as γ -gt Cre transgenic mice, we believe the conclusion should be similar to those obtained using Ksp Cre mouse.

Regarding the tubular segments in which beta-catenin palmitoylation affects kidney fibrosis, unfortunately, no studies have conclusively identified which segment of tubules as the main driver of renal fibrosis (PMID: 1942768; PMID: 22009250; PMID: 35788561). Damaged tubular epithelial cells (TECs) contribute directly to interstitial inflammation and fibrosis through various mechanisms (PMID:20436483; PMID: 32060481). As reported, β -catenin is essential for the development and maturation of multiple nephron segments in the mammalian kidney (PMID: 22021707; PMID:25647637). While β -catenin signaling remains suppressed in the adult healthy kidney, but is activated in the tubules in various types of CKD, including the unilateral ureteral obstruction (UUO), ischemia-reperfusion injury (IRI), angiotensin or adriamycin infusion, diabetic kidney disease, and age-related kidney dysfunction models (PMID: 15944336; PMID: 19297557; PMID: 31318148; PMID: 25855776; PMID: 28270411). Despite these findings, two questions remain unclear: First, in which tubular segments is β -catenin primarily increased during fibrosis? Second, in which tubular segments

does the increase in β -catenin play a major pro-fibrotic role? Nonetheless, it is clear that β -catenin is extensively increased in the renal tubules during fibrosis. We speculate that downregulation of DHHC9 in tubule cells from each segment leads to the reduction of β -catenin palmitoylation at this segment, which results in increased β -catenin accumulation and promotes renal fibrosis. Although we cannot conclusively determine which segmental tubular cell β -catenin palmitoylation occurs that primarily influences the outcome of renal fibrosis, it is worthy of further study in our future plan.

2. Would a mouse bearing a beta catenin C300S mutation exhibit enhanced kidney injury, assuming the mice are viable? This would be a key reagent to test the authors central hypothesis.

Response: Thank you for bringing up this point. In this study, we conducted in situ injections of adeno-associated virus (AAV) containing either β -catenin WT or C300S in mouse kidneys (Supplementary Fig. 10a, g). The results showed that mice injected with C300S AAV exhibited more severe renal fibrosis compared to those injected with WT β -catenin after UUO or IRI (Supplementary Fig. 10b-f, h-l).

3. Quantification of key experiments in Figs 5-7, and in supplemental figures should be provided, as you are only proving data from representative experiments. For example, I would like to see quantification of the data presented in Fig 5H, where you present data suggesting that C300 on beta catenin is a key palmitoylation site. There should be reduced palmitoylation of beta catenin in cells treated with a DHHC9 siRNA (5E). I am not appreciating this when normalized to a loading control, although it may be reduced when normalized to total beta catenin in the representative figure. I am also not seeing changes in beta catenin levels in cells transfected with an APT1 siRNA (6G). In 6N, I do not appreciate differences in beta catenin expression across the conditions. Data for Fig 7Q and 7R should be quantified.

Response: We do appreciate the reviewer's comments. We did the quantitative

analysis as suggested (Fig. 5e, 5f, 5h, Fig. 6g and Supplementary Fig. 9g). We replaced a more representative image in Fig. 6n. The quantitative analysis results were added in Fig. 7q and 7r.

4. Which is more relevant in examining the effects of DHHC9 on fibrosis, total or phosphorylated beta catenin?

Response: We appreciate the comment raised by the reviewer. Our study showed that DHHC9 plays a crucial role in regulating total β -catenin abundance, which is more relevant with kidney fibrosis.

5. I am having difficulty reviewing your figures based on information provide in the figure legend.

Response: Thank you for bringing this to our attention. We have updated the figure legends in the revised manuscript.

6. While your data provide strong evidence for a role of DHHC9 in palmitoylating beta catenin and targeting it for degradation, I would be cautious in ascribing all the affects you are seeing to DHHC9. DHHCs can assemble as heterooligomers, and it is possible that DHHC9 assembles with other DHHCs in targeting beta catenin for palmitoylation. Also, DHHC9 facilitates palmitoylation of multiple proteins and its possible that palmitoylation of other proteins could influence fibrosis in kidney injury models.

Response: We appreciate and agree with the reviewer's important point. Regarding the other DHHCs assembling with DHHC9 in targeting beta-catenin for palmitoylation, in this study, we found that DHHC9 protein can directly bind to β -catenin (Supplementary Fig. 9c), and more importantly, the in vitro PAT assays showed that DHHC9 protein alone can directly catalyze β -catenin palmitoylation (Fig. 5g). Therefore, these data suggest that DHHC9 alone can catalyze the palmitoylation of β -catenin, but whether the presence of other DHHCs can accelerate or limit this process needs more investigation.

Regarding whether DHHC9 can regulate the substrates other than β -catenin involved in renal fibrosis, the previous studies have shown that RAS proteins are

DHHC9 substrates (PMID: 16000296; PMID: 24127608; PMID: 17519897). DHHC9-catalyzed palmitoylation may regulate Ras membrane localization and function (PMID: 33816563; PMID: 23356262; PMID: 25158650). Meanwhile, increased membrane localization of RAS proteins enhances TGF-induced extracellular matrix production (PMID: 32049037, PMID: 31915377). In addition, DHHC9 can palmitoylate Glut1 to facilitate its membrane localization and promote its glucose uptake (PMID: 34620861). In this study we found that DHHC9 overexpression effectively alleviated fibrosis. Furthermore, many other substrates of DHHC9 such as IL-20R, CREB, IL-4, and ARG1 that may be related to fibrosis were found by searching the protein interaction data websites (<https://thebiogrid.org/119302>) (Supplementary Fig.13). We will work on these molecules in our future research plan.

7. For the Click it assay in mice, you mention that mice received an IP injection 9 after UUO or on day 20 after IRI. You also mention - at 6 hours after the fourth injection ... However, you do not mention when the three additional injections were administered.

Response: We apologize for the confusion and have revised the manuscript to clarify the administration of the additional injections. We typically administered IP injections of Azide-labeled palmitic acid at 8 a.m. on the day prior to the experimental endpoint, followed by subsequent injections at 6-hour intervals (2 p.m., 8 p.m., and 2 a.m. the following day), with the mice being sacrificed at 8 a.m. on the day of the endpoint.

REVIEWER COMMENTS

Reviewer #1 (Remarks to the Author):

Authors have addressed my concerns. No further question.

Reviewer #2 (Remarks to the Author):

As I mentioned in my previous review, this work represents a thoughtful and very thorough analysis of a novel mechanism that regulates beta catenin expression and fibrosis in the setting of kidney injury (UUO and IRI). A major concern was the Ksp Cadherin cre mouse to generate tubular specific APT1, DHHC 9 and beta catenin knockouts. With this system Cre is primarily expressed in distal nephron segments, as the authors acknowledge. As a major site of DHHC 9 (and IRI injury) is the proximal tubule, I doubt that there has been an efficient proximal tubular knockout of these three genes. The authors should provide evidence that there has been efficient knockout of DHHC9, APT1 and beta catenin in the proximal tubule, examining either protein (immunofluorescence) or message expression (RNAscope) in parallel with proximal tubular markers. This is a small part, but important part of the manuscript. If I am correct and that there has been an inefficient proximal tubular knockout, this need to be acknowledged.

My other concerns has been addressed.

Reviewer #1 (Remarks to the Author):

Authors have addressed my concerns. No further question.

Response: Thank you very much for your positive feedback on our study.

Reviewer #2 (Remarks to the Author):

As I mentioned in my previous review, this work represents a thoughtful and very thorough analysis of a novel mechanism that regulates beta catenin expression and fibrosis in the setting of kidney injury (UUO and IRI). A major concern was the Ksp Cadherin cre mouse to generate tubular specific APT1, DHHC 9 and beta catenin knockouts. With this system Cre is primarily expressed in distal nephron segments, as the authors acknowledge. As a major site of DHHC 9 (and IRI injury) is the proximal tubule, I doubt that there has been an efficient proximal tubular knockout of these three genes. The authors should provide evidence that there has been efficient knockout of DHHC9, APT1 and beta catenin in the proximal tubule, examining either protein (immunofluorescence) or message expression (RNAscope) in parallel with proximal tubular markers. This is a small part, but important part of the manuscript. If I am correct and that there has been an inefficient proximal tubular knockout, this need to be acknowledged.

Response: We sincerely appreciate the reviewer's thoughtful assessment of the potential significance of our research. One concern raised pertains to the utilization of Ksp Cadherin cre mice in our investigation, given that the primary site of DHHC 9 (and IRI injury) is the proximal tubule. In order to address this concern, we undertook co-staining of kidney tissues with an anti-DHHC9 antibody alongside PHA-E or PNA, enabling the identification of the deletion efficiency of the target gene in the proximal and distal tubules, respectively. In congruence with the immunohistochemical staining outcomes (Figure 2g, 2i, and 2l), DHHC9 expression was observed in both proximal and distal tubules of wild type mice,

while this expression was abrogated in the tubules with varying degrees (~80% in distal tubules, ~20% in proximal tubules) in the knockouts (Supplementary Fig. 16a). The expression and the knockout efficiency for beta-catenin in the proximal and distal tubules were similar to DHHC9 (Supplementary Fig. 16b).

Under physiological conditions, APT1 exhibited prominent expression within the distal tubules and were virtually undetectable within the proximal tubules of wild type kidneys. The knockout efficiency for APT1 in distal tubules was approximately 80% in the knockout mice (Supplementary Fig. 16c).

We concur with the reviewer's point that the knockout efficiency of target genes in the distal tubules significantly surpasses that in the proximal tubules through the use of Ksp cre transgenic mice. However, there are pertinent aspects to consider. First, while the proximal tubule is the principal site affected by acute ischemia-reperfusion injury, it's worth noting that the distal tubule also experiences a degree of damage (PMID: 37479698; PMID: 36798850). Ksp cre demonstrates selective expression across various renal tubular segments in mice (PMID: 36669643; PMID: 29382757), with multiple published studies utilizing Ksp cre transgenic mice to establish models of tubular cell-specific deletion in acute kidney injury scenarios (PMID: 36859428; PMID: 37460623; PMID: 34774556, et al). Second, the primary aim of our study was not focused on acute kidney injury per se. Rather, our study aimed to delve into the contributions of tubular cell DHHC9, APT, and β -catenin in renal fibrosis induced by UUO or IRI, encompassing both proximal and distal tubules, as documented in existing literature (UUO: PMID: 12631121, PMID: 22535799; IRI: PMID: 26701981, PMID: 28978534, PMID: 33440212). Last, it's essential to acknowledge that due to the relatively lower efficacy of knockout in the proximal tubules, the role of DHHC9 in the regulation of kidney fibrosis might potentially be underestimated, as opposed to overestimated, within the scope of this study using these transgenic mice. We have duly acknowledged this limitation and its implications in the

discussion section of the revised manuscript, as suggested by the reviewer. Your consideration of these nuances is greatly appreciated.